# Flexible mapping of abstract domains by grid cells via self-supervised extraction and projection of generalized velocity signals

**Abhiram Iyer**[1, 3, 4], **Sarthak Chandra**[2, 3], **Sugandha Sharma**[2, 3], and **Ila Fiete**[2, 3, 4]

[1]Department of Electrical Engineering and Computer Science, MIT, Cambridge, MA
[2]Department of Brain and Cognitive Sciences, MIT, Cambridge, MA
[3]McGovern Institute for Brain Research, MIT, Cambridge, MA
[4]K. Lisa Yang Integrative Computational Neuroscience (ICoN), MIT, Cambridge, MA ,
{abiyer,sarthakc,susharma,fiete}@mit.edu

## Abstract

Grid cells in the medial entorhinal cortex create remarkable periodic maps of explored space during navigation. Recent studies show that they form similar maps of abstract cognitive spaces. Examples of such abstract environments include auditory tone sequences in which the pitch is continuously varied or images in which abstract features are continuously deformed (e.g., a cartoon bird whose legs stretch and shrink). Here, we hypothesize that the brain generalizes how it maps spatial domains to mapping abstract spaces. To sidestep the computational cost of learning representations for each high-dimensional sensory input, the brain extracts self-consistent, low-dimensional descriptions of displacements across abstract spaces, leveraging the spatial velocity integration of grid cells to efficiently build maps of different domains. Our neural network model for abstract velocity extraction factorizes the content of these abstract domains from displacements within the domains to generate content-independent and self-consistent, low-dimensional velocity estimates. Crucially, it uses a self-supervised geometric consistency constraint that requires displacements along closed loop trajectories to sum to zero, an integration that is itself performed by the downstream grid cell circuit over learning. This process results in high fidelity estimates of velocities and allowed transitions in abstract domains, a crucial prerequisite for efficient map generation in these high-dimensional environments. We also show how our method outperforms traditional dimensionality reduction and deep-learning based motion extraction networks on the same set of tasks. This is the first neural network model to explain how grid cells can flexibly represent different abstract spaces and makes the novel prediction that they should do so while maintaining their population correlation and manifold structure across domains. Fundamentally, our model sheds light on the mechanistic origins of cognitive flexibility and transfer of representations across vastly different domains in brains, providing a potential self-supervised learning (SSL) framework for leveraging similar ideas in transfer learning and data-efficient generalization in machine learning and robotics.

## 1 Introduction

Grid cells in the medial entorhinal cortex are of paramount importance for navigating and representing spatial domains. Interestingly, a series of recent experiments have shown that the brain still uses the same cells to represent non-spatial environments that are continuously traversed. These include free

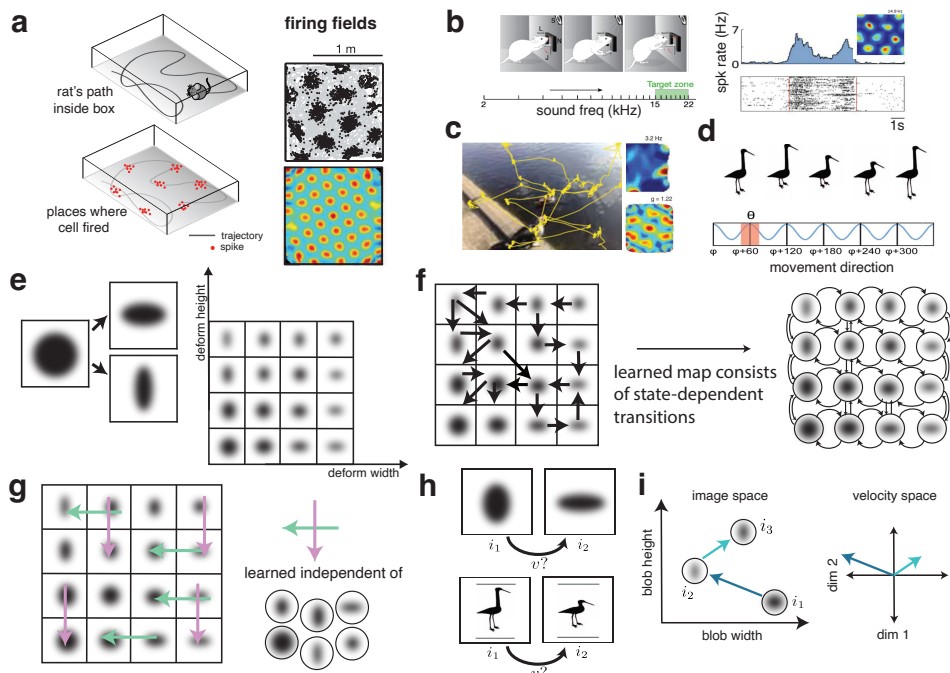

Figure 1: **Conceptual understanding of learning velocities in abstract cognitive environments.**
**a.** Grid cells generate hexagonal tuning in spatial navigation experiments (illustration borrowed from [1]). **b.** Similar grid-like tuning has been found in auditory frequency space in rodents [2], (**c.**) in visual space in monkeys [3], and (**d.**) in an abstract cartoon bird space in human experiments. **e.** Example of an abstract, non-spatial domain called 'Stretchy Blob', a 2D Gaussian that can either stretch or shrink along two axes. **f.** SR, CSCG, and TEM [4–7] learn transition structures of this cognitive domain by traversing it. These models require one to build a set of representations for these encountered states as well as structures enabling transitions between them. **g.** Our approach learns a self-consistent movement (velocity) signal that is independent of the states traversed and encodes the global transition structure of the environment in a minimally low-dimensional representation. **h.** How can the brain initially extract a general notion of velocity within abstract, non-spatial domains? Solving this important prerequisite challenge can show how grid cells flexibly map and organize various abstract spaces. **i.** The actual movement signals in this particular example are low-dimensional. These velocities are independent from the states they connect.

viewing of naturalistic images, listening to and modifying a sound changing in pitch, navigating a conceptual 'Stretchy Bird' space (images of a bird that stretch or shrink via joystick input), or even traversing an 'odor' space among many others [8–14, 3, 15, 2] (Fig. 1a-d).

How does the brain transfer its ability to represent space to these non-spatial environments? More specifically, how does the brain infer a metric layout in these abstract domains? Answering this question involves understanding how the brain perceives and processes structure in different domains and modalities, and integrates them into coherent cognitive maps.

How the brain maps abstract cognitive environments has been a question of intense interest, with several proposed approaches [4, 5, 7, 6]. These works propose that when the brain learns the transition structure of a cognitive domain (such as the 'Stretchy Blob' space in Fig. 1e), it simultaneously builds a set of representations for these states as well as structures enabling transitions between them, Fig. 1f.

Our approach investigates how the brain exploits an existing scaffold of structured states, provided by grid cells, to represent new cognitive domains by projecting them onto the invariant grid coding space. We hypothesize that this projection involves anchoring a state in the cognitive domain to a grid phase and then extracting a self-consistent movement signal to measure displacements in the cognitive domain (Fig. 1g). This abstract velocity signal serves as the input to the grid cell network, updating grid phases and encoding transitions between high-dimensional states using the existing

low-dimensional transition mechanism within the grid cell circuit. Crucially, rather than building a *new set of continuous stable states* for each explored cognitive domain — a process that is both difficult and slow — this mechanism allows for the *efficient reuse of a canonical scaffold of cognitive states* for the memory and coding of continuous variables.

In spatial domains, continuous attractor models of the grid cell circuit [16] efficiently reuse prestructured grid cell states to encode transitions between high-dimensional states. However, using such models to map non-spatial, abstract domains (and by extension, explaining how grid cells can be efficiently reused) requires the extraction of faithful representations of velocity in these domains. Thus, the fundamental, prerequisite challenge lies in extracting a general notion of a minimal dimensional velocity signal from high-dimensional, time-varying data from various abstract domains, Fig. 1h. Crucially, the representations of this extracted velocity must be independent of the specific states, and must be self-consistent with a net-zero velocity corresponding to a net identity transformation in the abstract space. These constraints point towards a self-supervised learning paradigm, wherein neither the coordinate system of the space nor the dimensionality of the estimated velocity are known a priori.

The following are the key contributions of the paper:

- **First neural network model for abstract velocity extraction.** We present the first neural network model explaining how grid cells flexibly represent different abstract spaces through providing a learning framework for velocity estimation in arbitrary spaces. This reuse of grid cells across domains leads to cognitive flexibility in mapping spaces, providing insights towards transfer learning and data-efficient generalization in machine learning and robotics.

- **State-independent velocity extraction.** Our framework for abstract velocity extraction generates state-independent, self-consistent low-dimensional velocity estimates by separating the content of abstract domains from their displacements (Fig. 1g,i).

- **Self-supervised geometric consistency constraint.** We show how consistent metric representations of velocity can be learned through a self-supervised geometric consistency constraint requiring displacements along closed loop trajectories to sum to zero — an operation facilitated by the downstream grid cell circuit.

- **Superior dimensionality reduction performance.** Our method surpasses traditional dimensionality reduction and deep learning-based motion extraction networks, specifically in environments characterized by underlying low-dimensional transitions and motions between states.

- **Preservation of cell-cell relationships.** The model predicts that cell-cell relationships between grid cells are preserved *across* different spatial and non-spatial domains. For example, if two grid cells are co-active in a spatial task, they should remain co-active in a non-spatial task. This prediction critically relies on the same continuous attractor-based grid module performing integration across domains — a capability that can only be realized through extraction of a state-independent velocity from abstract domains. This prediction aligns with the observed invariance of internal neural representations across different spatial environments and brain states [17–23]. This forms a testable hypothesis for future experimental studies on neural representations in abstract cognitive domains.

## 1.1 Related work

Our work on learning velocities and transition operators in abstract cognitive spaces can be compared to two key areas: neuroscience models of spatial mapping and dimensionality reduction models for high-dimensional data. We first discuss cognitive space mapping models and then relate them to dimensionality reduction work.

The Tolman-Eichenbaum Machine (TEM) [7] learns a map of a 'cognitive domain' via a recurrent neural network, predicting sensory observations based on given actions. However, TEM does not infer transition structures or affordances, relying instead on predefined actions at each time step. When presented with a new environment, TEM requires re-learning these affordances along with grid cell-like representations from scratch. Similarly, the successor representation (SR) [4, 5] predicts future states as a weighted sum of expected future occurrences and uses an underlying discrete action space. State-action variants of SR are dependent on these discrete action inputs to construct

cognitive maps. Clone-structured cognitive graphs (CSCG) [6] learn hidden Markov models of sensory representations from observational inputs without prior domain structure assumptions but are limited to only discrete domains. By construction, these models are not guaranteed to preserve cell-cell correlations across domains and modalities and are unable to efficiently reuse the prestructured states and transition operators provided by grid cells.

More generally, these models posit that the brain simultaneously learns representations of the external states *and* the transition structures to form cognitive maps. Here, we propose that approaching the cognitive mapping problem from a velocity-extraction-first perspective offers key benefits. By learning a low-dimensional velocity that captures transitions between external inputs, mapping the environment to reusable grid cell states can be performed simply by a continuous attractor network, instead of something more complex like TEM. More generally, via our self-supervised learning framework that infers velocity solely from sensory inputs in continuous domains, we can entirely avoid the computationally expensive task of representation and transition learning through capturing the minimal low-dimensional structure of these abstract observations.

Our work thus bridges the tasks of building cognitive maps of abstract domains and dimensionality reduction. Previous dimensionality reduction methods [24–29] focus on preserving proximity in high-dimensional spaces on low-dimensional manifolds but do not take advantage of the structure imposed on the tangent spaces of these manifold (i.e., velocity spaces) through transitions between states. As a result, while these methods learn low-dimensional mappings, they often fail to reproduce the underlying metric space of abstract domains. Unlike these methods, our framework learns a global velocity operator applicable to any input within the environment's state manifold. Our work also relates to the task of motion decomposition and next-frame prediction of temporally varying inputs [30–35]. However, these works have not attempted to decipher a minimally low dimensional description of velocity, making their learned representations unsuitable for grid cells to effectively map observed environments.

## 2 Self-supervision for velocity extraction

Learning action primitives purely through self-supervision, without access to true velocities in the space, requires careful consideration of the data generation process, the loss terms to extract meaningful neural representations, and the chosen neural network architecture. We will discuss each of these aspects in detail.

### 2.1 Task setup

The initial step in our methodology involves the construction of tasks designed to facilitate the model's inference of the lowest-dimensional representations of velocity within cognitive domains.

We revisit the 'Stretchy Blob' environment that we previously introduced. This environment can be procedurally generated where state transitions are characterized by changes in the blob's width and height. As seen in Fig. 2a, given two images, $i_1$ and $i_2$, we aim to learn a function $f$ that infers a low-dimensional velocity $\hat{v}$ representing the transition from $i_1$ to $i_2$ (Fig. 2b).

To ensure that $\hat{v}$ is a useful and faithful metric representation of the transition between two inputs, we demand that it satisfy two properties.

First, the estimated transition velocity $\hat{v}$ can be used to transform a given input, i.e., there exists a function $g$ that uses $\hat{v}$ to transform $i_1$ to predict $i_2$. In practice, to ensure that $f$ does not merely memorize image features of $i_2$, we demand that $g$ predict an unseen $i_3$ obtained by transforming $i_2$ using the same transition $\hat{v}$. More generally, this ensures that the estimates of displacements, $\hat{v}$, in this space must be independent of the route taken, ensuring both path and state independence. We refer to this requirement as 'next-state prediction'.

Second, the estimated velocities in the abstract space, $\hat{v}$, must be geometrically self-consistent, i.e., the start and end point of a trajectory in the abstract space are identical if, and only if, the sum of estimated velocities is zero (Fig. 2c). (Note that this also ensures that the identity transformation should be represented by the zero-vector.) We refer to this requirement as 'loop-closure'. This summation of velocities must be performed by a neural integrator, such as grid cells, which results in a calibrated velocity input for grid cells to then themselves use to map out the input abstract domain.

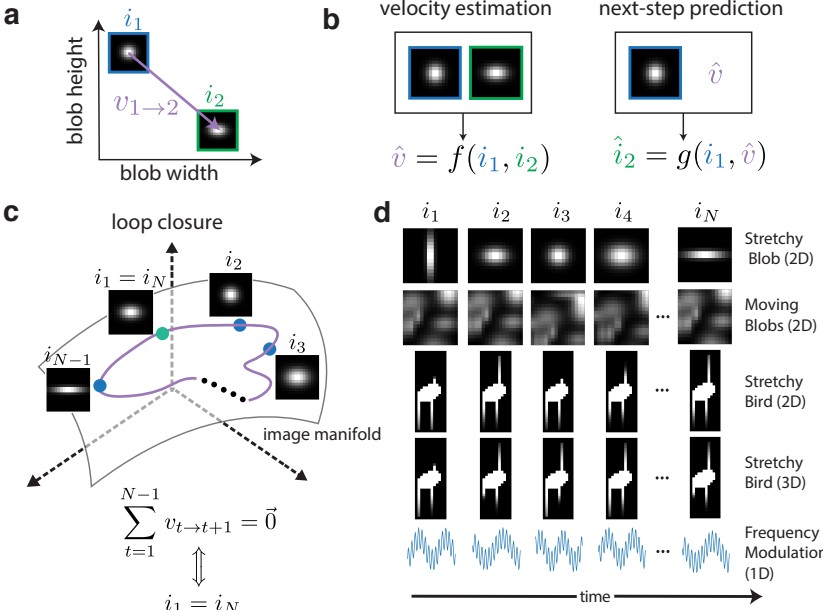

Figure 2: **Task and fundamental problem setup. a.** Two states in the 'Stretchy Blob' task that have a low-dimensional velocity representing the transition between them. **b.** If two consecutive images in this space $i_1$ and $i_2$ are separated by a velocity $v_{1\rightarrow2}$, can we learn a function $f$ that estimates this velocity and a function $g$ that 'applies' this quantity to $i_1$ to predict $i_2$? **c.** The key self-consistency constraint we introduce called 'loop-closure': estimated velocities along a closed-loop trajectory must sum to zero. This computation is performed by a neural integrator, such as grid cells. **d.** Our various procedurally generated abstract cognitive domains: Stretchy Blob (2D), Moving Blobs (2D), Stretchy Bird (2D), Stretchy Bird (3D), and Frequency Modulation (1D).

These self-consistency constraints on $\hat{v}$ suggests a *self-supervised learning paradigm, obviating the need for a specific coordinate system within the explored abstract space*.

To rigorously evaluate our system, we procedurally generate random trajectories across five different "abstract cognitive domains." These trajectories are generated by starting from an initial random state on the image manifold and then taking random velocities to determine the subsequent states on the same manifold. We assume for simplicity that states are unique at each point in space within these domains. We visualize a randomly sampled trajectory of each of these domains in Fig. 2d:

1. A 2D Stretchy Blob environment (discussed above) where a blob in the center of the visual field stretches or shrinks in height and/or width.

2. A 2D Stretchy Bird environment where a bird's legs and neck stretch and shrink. This environment is specifically constructed to mimic its experimental counterpart [15].

3. A 3D variant of Stretchy Bird where the two bird legs can independently transform.

4. A 2D Moving Blobs environment where a set of Gaussian blobs uniformly translate, moving in and out of the visual field.

5. A 1D Frequency Modulation task that emulates the experimental sound modulation task created by [2] with a sum of sine waves uniformly changing in frequency.

Detailed information regarding the generated data is provided in SI Sec. C.

## 2.2 Encoder-Decoder architecture

We construct an encoder-decoder architecture, wherein the encoder $f$ and decoder $g$ are both multi-layer perceptrons (MLPs) (Fig. 3a). The encoder $f$ processes two adjacent inputs $i_t$ and $i_{t+1}$ from any of our procedurally generated trajectories and maps them to a low-dimensional velocity space. It

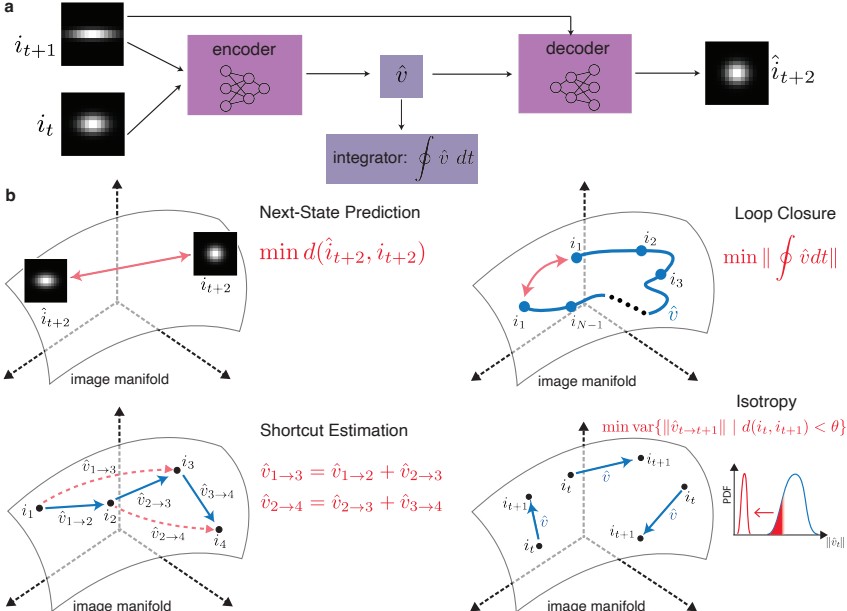

Figure 3: **Self-supervised learning framework. a.** Model diagram consisting of an encoder, decoder, and an integrator which acts on the low-dimensional velocity latent space. The model takes in two consecutive input frames and predicts an unseen frame with the learned velocity. **b.** Our various self-supervised loss terms. The two critical loss terms (*top*) are 'next-state prediction' and 'loop-closure'. The auxiliary loss terms (*bottom*) which further refine the solution space are 'shortcut estimation' and 'isotropy'.

is important to note that the low-dimensional velocity signal is not known a priori. The decoder $g$ then upsamples this latent velocity representation and combines it with the last input to generate the subsequent $\hat{i}_{t+2}$ which remains unseen to the model.

A single sample in a batch of data for this model comprises a trajectory of states $\mathcal{T} = \{i_t \,|\, 1 \leq t \leq N\}$ and velocities $\mathcal{V} = \{v_{t \to t+1} \,|\, 1 \leq t \leq N - 1\}$ (individual velocities abbreviated as $v_t$ for ease). The model is trained through triplets of frames of the input trajectory, using $i_t$ and $i_{t+1}$ to predict $i_{t+2}$ for $1 \leq t \leq N - 1$. As discussed above, we train our models on triplets of states solely to ensure that the encoder does not memorize features of the image to be predicted. Training on pairs of images instead of triplets does not affect any of our results (SI Sec. B.2).

Detailed experimental procedures regarding the training of this model across various constructed domains are provided in SI Sec. C. We note that the same architectural motif was employed for training in all our experiments.

## 2.3 Loss functions

We formalize our two requirements of the extracted low-dimensional velocity signal into two critical loss terms, a next-state prediction loss, and a loop-closure loss. These losses from the core of our self-supervised learning framework. To further refine the solution space, we also employ auxiliary losses in addition to our primary constraints. Our loss terms are visualized in Figure 3b and are described in detail in SI Sec. C.

- **Next-State Prediction Loss.** We quantify this based on the difference between the next-state prediction $\hat{i}_{t+2}$ and the ground-truth frame $i_{t+2}$. This loss term operates on individual samples of the generated trajectory, and ensures decodability of the estimated velocity $\hat{v}$.

- **Loop-Closure Loss.** We quantify this as a norm of the sum of velocities along a closed loop trajectory $\mathcal{T}$, i.e., the model must produce velocity estimates such that $\sum_{\hat{v} \in \hat{\mathcal{V}}} \hat{v} = \vec{0}$. The error signal for this loss operates at the scale of the entire generated trajectory. For convenience, we construct all trajectories of our training data as random loops in the

considered abstract spaces, such that the start and end state of trajectories are identical. See SI Sec. B.2 for training data that is not solely loops.

- **Shortcut Estimation Loss.** The first of our two auxiliary loss terms is a shortcut estimation loss, which further tests the generalization ability of our decoder $g$. From $i_{t+1}$, we predict future states $i_{t+3}$ or $i_{t+4}$ by directly modifying $\hat{v}$. Specifically, if $\hat{v}_{2\to3}$ is inferred from $i_{t+2}$ to $i_{t+3}$ and $\hat{v}_{3\to4}$ is inferred from $i_{t+3}$ to $i_{t+4}$, then $\hat{\hat{i}}_{t+4}$ should be $\hat{v}_{2\to3} + \hat{v}_{3\to4}$ away from $i_{t+2}$. This loss is important for further refining our velocity estimates and ensuring their validity during generalization.

- **Isotropy Loss.** The loss terms considered so far do not ensure isotropy in the inferred velocity space, allowing differential scaling factors for transformations in different input space directions. To induce isotropy, we introduce an auxiliary isotropy loss term that acts on the norm of the velocities, independent of direction. Since we don't assume access to the global velocity distribution in the training data, the isotropy loss is applied only near zero velocity. In particular, we minimize the variance of $\{\|\hat{v}_{t\to t+1}\| \mid d(i_t, i_{t+1}) < \theta\}$, where $d(i_t, i_{t+1})$ is a similarity metric in the input image space, and $\theta$ is some small threshold. In practice, we use $1-$ cosine similarity as our distance metric $d$.

There is no direct supervision signal regressing the model outputs onto a known distribution of velocities; instead, the velocities are extracted and inferred automatically. Further, we do not a priori assume knowledge of the dimensionality of the underlying transition structure of the training domains.

Regardless of the training environment, the relative weighting ratio of the two critical loss terms remains consistent, with loop-closure loss always weighted ten times higher than next-state prediction loss. To stabilize the training of models with three-dimensional latent spaces, we include a small $L_1$ regularization during training. All models are evaluated on an unseen testing dataset consisting of random walks within the domain. To prevent overfitting, all models are deliberately underparameterized relative to the training dataset. We conduct ablation studies on our loss terms, which can be found in SI Sec. B.1.

## 3 Experimental results

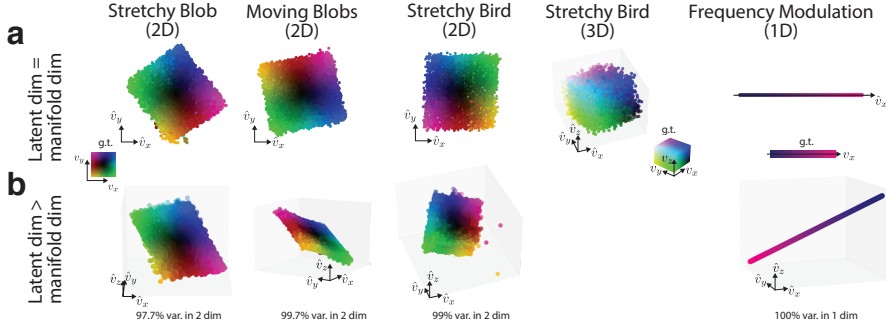

Figure 4: **Model results. a.** Our model produces low-dimensional velocity latents that are similar to the ground truth (g.t.) distribution without knowing this distribution across a variety of cognitive environments. **b.** In cases where the model's latent dimensionality is higher than the intrinsic velocity dimensionality of the environment, our model still identifies the lowest-dimensional representations embedded in higher-dimensional space.

### 3.1 Single learning framework infers geometrically consistent representations of velocity across cognitive domains

For each abstract environment, we compare the model-inferred velocity representations (Fig. 4a) to the true distribution of velocities. Note that the self-supervised framework does not result in an exact identity mapping between the ground truth velocities and the model outputs — it suffices for the obtained output to be a linear transformation of the ground-truth velocity space. Correspondingly, we

| Task | Our Model | MCNet | Autoencoder | PCA | Isomap | UMAP |
|---|---|---|---|---|---|---|
| Stretchy Blob (2D) | **0.05 ± 0.01** | 1.95 ± 0.14 | $(1.59 ± 3.40) ×10^3$ | 0.63 | 0.42 | 0.79 |
| Stretchy Bird (2D) | **0.07 ± 0.03** | 2.01 ± 0.01 | 20.90 ± 38.04 | 0.21 | 0.36 | 0.46 |
| Stretchy Bird (3D) | **0.07 ± 0.02** | 2.96 ± 0.10 | $(2.66 ± 5.08) ×10^2$ | 0.31 | 0.64 | 1.05 |
| Moving Blobs (2D) | **0.02 ± 0.01** | 2.00 ± 0.01 | 2.03 ± 0.05 | 1.94 | 0.66 | 0.62 |
| Freq. Modulation (1D) | **0.02 ± 0.02** | 2.01 ± 0.01 | 2.00 ± 0.01 | 1.97 | 2.00 | 2.00 |

Table 1: **Mean and standard deviation of errors for various tasks and models.** Mean and standard deviation of errors are computed across 6 different runs for each experiment. Each run was seeded to ensure reproducibility. The 6 seeds were picked randomly and are the same seeds used across different experiments where multiple runs were run.

construct an error metric on the inferred velocities as the mean squared error in mapping the predicted velocities to the ground-truth via a single linear transformation, after removing a small number of outliers via the DBSCAN clustering algorithm [36]. More details about this error metric can be found in SI Sec. C.

In all cases, irrespective of the dimensionality of the input manifold space or the detailed statistics and structure of the environment states, we see that the inferred velocities are faithful metric representations of the ground-truth velocities, quantified in Table 1.

While we primarily consider cases where the latent dimensionality of the encoder output matches the underlying dimensionality of transitions in the input space, this is not a necessity. In Fig. 4b, we set up our framework to have latent dimensionalities larger than the true data manifold dimensionality. In all cases, the model outputs *automatically* occupy a subspace of dimensionality that corresponds to the actual input manifold transition space, with a PCA of the inferred velocities capturing greater than 97% of the variance within the correct number of dimensions (cf. Fig. 4b). Thus, our model can effectively identify the appropriate low-dimensional structure within the high-dimensional embedding space of the inputs. Results from the other synthetic cognitive domains can be found in SI Sec. A.

## 3.2 Comparison to existing dimensionality reduction methods

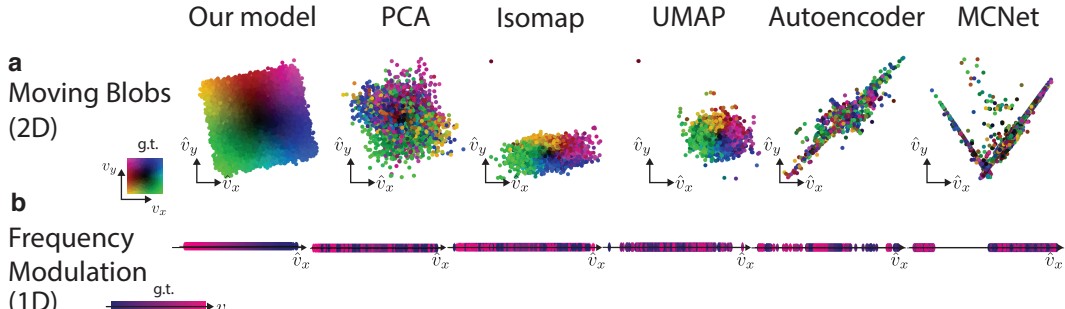

Figure 5: **Comparison to baselines.** We show our model comparisons to various dimensionality reduction and motion-prediction baselines in the **a.** 2D Moving Blobs and **b.** 1D Frequency Modulation tasks. Existing baselines cannot identify the low-dimensional velocity signals between arbitrary transitions in this space, even failing to do so in a simple one-dimensional domain. Meanwhile, our model produces results that closely match the true, underlying velocity distribution.

Our model estimates low-dimensional velocities between successive high-dimensional states. These velocities can then be integrated to determine a low-dimensional representation for each state. In this sense, it is possible to view our work as a dimensionality reduction method for continuously varying inputs. Traditional dimensionality reduction methods rely on the *statistics of distances between points* across an ensemble of states. In contrast, our approach finds a *structured tangent manifold* around each state that captures the low-dimensional transitions to successive states.

We can compare standard dimensionality reduction techniques such as PCA, Isomap, UMAP, and deep autoencoders with our method. To do so, we use these techniques to embed the inputs into the known

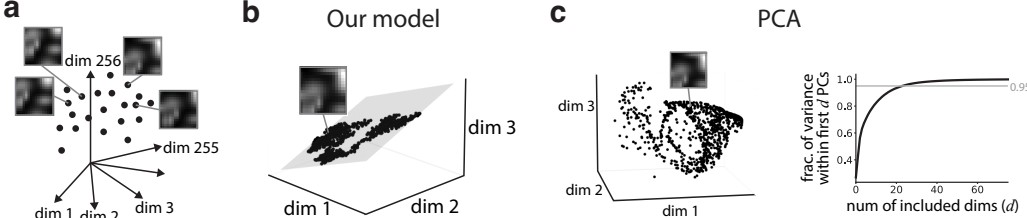

Figure 6: **Dimensionality reduction through our model, versus PCA. a.** Schematic of the raw input states from a random trajectory of the 2D Moving Blob task, showing the states as points in a 16x16 dimensional space. **b.** We estimate 3-dimensional velocities between states, and integrate these estimated velocities to obtain a low-dimensional representation of the initial input states. A 2-dimensional plane is shown in gray for perspective, demonstrating our model-produced low-dimensional representations are approximately in a 2D subspace. **c.** *Left*: Computing PCA on the same dataset shows representations occupying a volume in a 3-dimensional space. *Right*: Around 24 dimensions are required for PCA to capture 95% of the variance in the data, indicating that PCA is unable to find a low-dimensional space describing the dataset.

latent dimensionality of the data manifold, then compute an estimated low-dimensional velocity between states by taking the difference of their corresponding low-dimensional representations. Fig. 5 compares our model's estimated velocities with those produced by these standard baselines. Since our self-supervised framework uses a next-step prediction component, we also compared our results to MCNet [30] (a flexible deep network designed specifically for future frame prediction) and constrained the model to use low-dimensional representations of the transitions between frames. In all cases, our model significantly outperformed other baseline models (cf. Table 1) and produced velocities that were closely aligned with the true velocity distributions. Remarkably, for even one-dimensional manifolds embedded in high-dimensional spaces (as in our 1D Frequency Modulation task), existing dimensionality reduction techniques struggle to find a coherent representation for velocity and produce errors that are two orders of magnitude larger than our model.

We also examine dimensionality reduction through constructing representations of the original data via integrating the model velocity estimates, Fig. 6. For a sample trajectory through the high-dimensional states of the 2D Moving Blob environment, our models representation collapses onto a two-dimensional plane (consistent with the data, since the states can be minimally described through transitions in $\mathbb{R}^2$). PCA of the same set of states fails to capture this low-dimensional description, with $\sim 24$ principal components necessary to capture $> 95\%$ of the variance.

### 3.3 Model outputs allow reuse of grid cells in mapping abstract domains

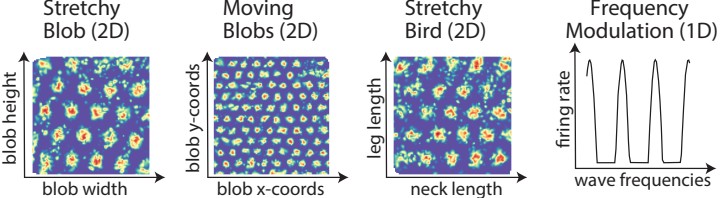

Figure 7: **Model outputs can be used to generate grid-like firing fields.** Our model's outputs can be used as input to a synthetic grid cell network across a variety of cognitive domains. We predict a clear hexagonal-like firing field when traversing these environments, illustrating how the grid cell circuit is crucial to building these cognitive maps.

We hypothesize that the brain can map abstract cognitive domains to the manifold of grid cell states by learning low-dimensional velocity representations of the input space. Extracting this low-dimensional signal allows the brain to reuse continuous attractor dynamics and path integration functionalities to stably represent and traverse along the manifold of inputs in the mapped domain.

To verify that the model inferred velocities are faithful enough for accurate path integration, we construct the tuning curve of a random neuron from a synthetic grid cell module (details of which are briefly described in SI Sec. C) that is fed these inferred velocities (up to a best-fit linear transformation; for simplicity we choose our transformation to map to aligned velocities across environments, leading to orientation-aligned grid tuning curves — this need not hold in experimentally observed tuning curves) as the path integration inputs. As seen in Fig. 7, our model results in hexagonal tuning curves in the abstract cognitive domains, consistent with previous work on grid representations [8–14, 3, 15, 2] (grid firing fields from baseline models explored in SI Sec. A). Through extracting faithful low-dimensional representations of velocities across abstract domains, the *same* continuous attractor-based grid modules can be used across tasks. As a result, if two grid cells fire in an overlapping (or non-overlapping) way in one mapped domain, they continue to be overlapping (or non-overlapping) in all other domains. Thus, cell-cell relationships are preserved between grid cells across abstract domains. This forms a testable hypothesis for future experimentation, that may be falsified if the brain were to use distinct, independent grid cell modules to organize information from different modalities, or if grid cells were not prestructured networks that function independently of the nature of the inputs.

We also note that while each grid cell module is a two-dimensional toroidal manifold, enabling integration of two-dimensional velocities, the grid cell system consisting of multiple modules can integrate velocities in higher dimensions[37]. Thus, higher dimensional velocities extracted by our model (e.g., 3D Stretchy Bird) do not pose a problem for integration by grid cells.

## 4   Discussion

Our work introduces the first neural network model that can infer velocities within abstract cognitive domains. This enables circuits like grid cells to encode transitions between high-dimensional sensory states through low-dimensional path integration, mapping sensory inputs to prestructured states instead of learning independent representations for each state. Our velocity extraction mechanism itself requires path-integration (via the 'loop-closure' loss), necessitating a grid-cell-like neural model, highlighting how grid cells form the foundation of mapping abstract spaces.

**Future Work.** Our research offers new perspectives for neuroscientists on the flexible utilization of grid cells to organize and map non-spatial domains. Future neuroscience research may test our hypothesis on the conservation of cell-cell relationships across cognitive domains and identify brain regions that generate velocity signals, along with their experimental signatures. Future machine learning research directions include scaling our framework for naturalistic environments, learning non-Euclidean spaces like family trees, and possibly using our framework to augment existing cognitive space-mapping models like TEM or CSCG.

**Limitations.** While our core SSL loss terms (next-state prediction and loop-closure) are biologically plausible and may align with sensory prediction error and neural integration, the auxiliary losses (shortcut and isotropy) are less biologically supported. Additionally, we assume velocity vectors in our latent space commute, which prevents them from directly representing tangent vectors in non-Euclidean spaces like a sphere. However, non-Euclidean spaces can be represented by embedding them in a higher-dimensional Euclidean space where velocities commute (e.g., a sphere embedded in three-dimensional space)[38, 39].

**Broader Impact to AI.** Our novel SSL framework can be applied for invertible dimensionality reduction (by virtue of a generative decoder which generates high-dimensional states corresponding to points in a low-dimensional latent) and manifold learning tasks. Our model significantly outperforms non-invertible dimensionality reduction baselines on datasets that contain relatively lower-dimensional transitions (suggesting applications to video data, for example). Our method also naturally lends itself to manifold alignment-related challenges, which is particularly effective when the data exhibits a small number of continuous modes of variability. Moreover, with a small number of "gluing" points, our method allows for building one-to-one correspondences between different domains. Our work also shows how a fixed integrator circuit can leverage common velocity representations to navigate between abstract spaces efficiently. For example, in a complex maze where learning action strategies is costly, our model maps transitions and actions to the grid-cell representational space, enabling strategies learned in a simpler, topologically similar space to be effectively applied to the complex domain — a relevant challenge in robotics.

## 5 Acknowledgements

We thank the McGovern Institute and the K. Lisa Yang Integrative Computational Neuroscience (ICoN) Center for supporting and funding this research. AI was supported by the K. Lisa Yang Integrative Computational Neuroscience (ICoN) Fellowship.

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

# A  Additional experiments

## A.1  Additional results across domains

Fig. 8 includes further experiments of our framework across different domains and also compare them to more baselines. Fig. 9 visualizes the grid firing fields using our model outputs as input to a synthetic grid cell network and compares them to the same baselines. Our model produces the most faithful representations of velocity across various domains.

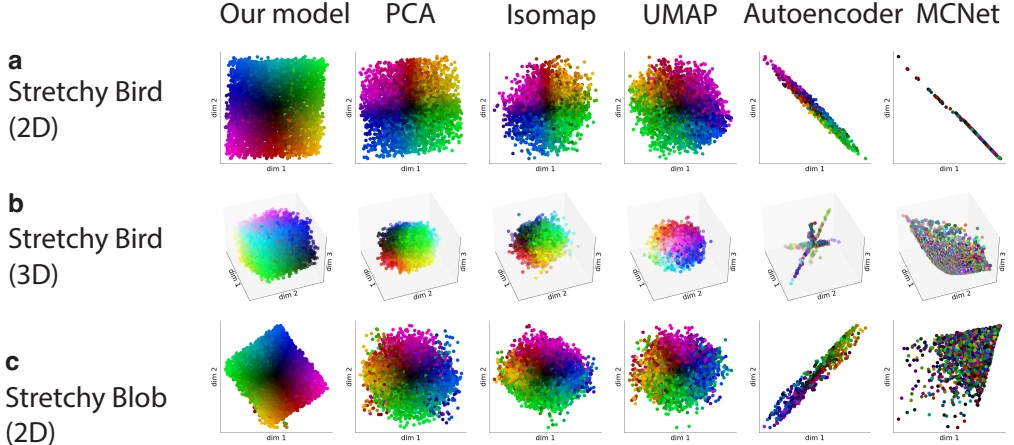

Figure 8: **Continued, comparison to baselines. a.** Model inferred velocity space in the 2D Stretchy Bird environment compared with baselines. **b.** Model inferred velocity space in the 3D Stretchy Bird environment compared with baselines. **c.** Model inferred velocity space in the 2D Stretchy Blob environment compared with baselines.

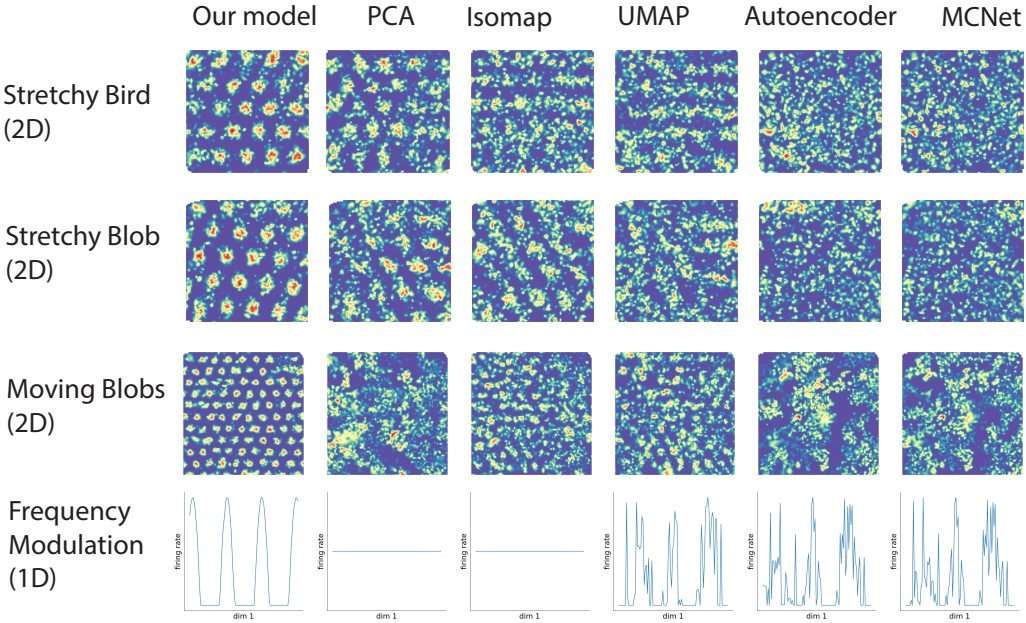

Figure 9: **Can baseline models produce faithful representations of velocity?** We pipe the outputs of our baselines into the same synthetic grid cell network to observe grid firing rates along a trajectory. Qualitatively, our model produces the most hexagonal grid firing field in comparison to other baselines.

### A.2 Decoder implicitly learns boundaries of training data manifold

While the encoder is state-independent and infers a generalized notion of velocity between any two high-dimensional states within our abstract domains, the decoder is state dependent by construction as it outputs a predicted state given a state and velocity. We investigate how the decoder performs at boundaries in the 2D Stretchy Bird environment, as illustrated in Fig. 10. We find that the decoder implicitly understands the boundaries of the training data's underlying manifold. That is, once a velocity produces a bird state that is unseen in the training distribution, i.e. a bird that cannot further shrink its legs or extend its neck, further transitions in the same direction do not produce any changes in the predicted state. Thus, the decoder understands the boundaries observed in the training data.

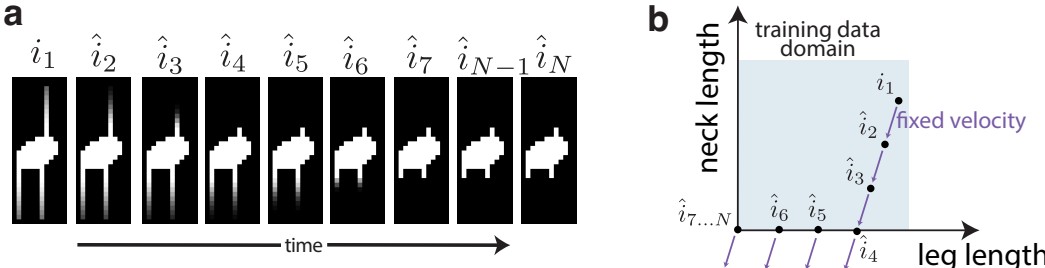

Figure 10: **Decoder respects boundaries inferred from the extent of the training data. a.** Starting from a random point in the 2D Stretchy Bird environment, we use the decoder to estimate the state after applying a velocity $v$. We then iteratively apply this same velocity $v$ to the generated state estimate from the previous step, generating a series of states obtained by traversing the environment while following this fixed velocity. We find that once the neck and leg length have maximally shrunk to the extent observed in the training data, the decoder arrives at a fixed point. Thus, the decoder performs state-dependent transformations: after reaching an inferred boundary of the training data, further velocities along the same direction do not continue to transform the data along that dimension.

## B Ablation studies

### B.1 Loss Ablations

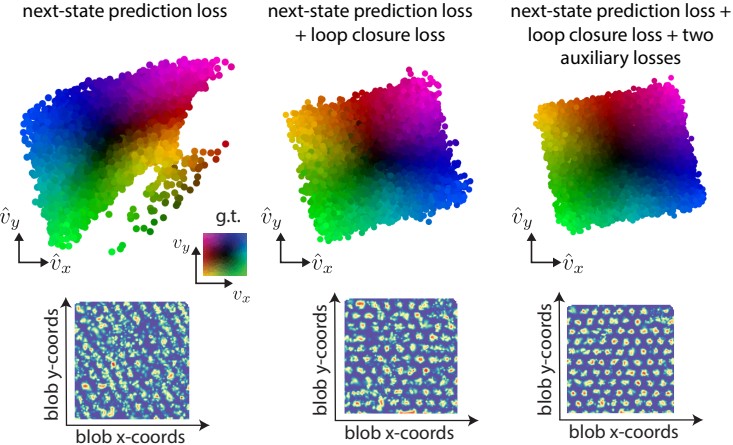

Figure 11: **Loss Ablation studies.** Loss ablation studies in the 2D Moving Blobs environment show that most of velocity estimation learning process comes from our two critical loss terms, 'next-state prediction' and 'loop-closure'. Further loss terms refine the solution space. Grid cell firing rates show that model generated velocities are faithful representations of the true, underlying velocity distribution.

We ablate our various loss terms in order to identify which losses are critical for faithful velocity extraction. As seen in Fig. 11 *left*, if we are interested in extracting some nonlinear low-dimensional

velocity between states in any abstract domain, the next-step prediction loss is enough (representation error of $0.134$). However, since grid cells are doing path integration through linear addition of velocity vectors, constraining our latent space to also be a linear function of the true velocity signals is essential. Thus, to demand that the estimated velocities be linear functions of the ground truth velocities, we require the loop-closure loss, the addition of which is visualized in Fig. 11 *middle* (representation error of $0.044$). If we require further refinement and guaranteed isotropic representations of velocities, we include our isotropy and shortcut losses, the addition of which is visualized in Fig. 11 *right* (representation error of $0.02$). Examining the grid cell tuning curve within this domain reveals that using only the next-step prediction loss results in non-hexagonal firing fields. In contrast, incorporating the loop-closure loss results in hexagonal tuning curves, with further refinement achieved by adding auxiliary losses.

For completeness, we prove that the loop-closure loss exacts a strong linearity constraint on its inputs.

**Proof:** Let the function $e : v_{t \to t+1} \to \hat{v}_{t \to t+1}$ represent the mapping from the true velocity to the model estimated one. The loop-closure loss, applied on trajectories that form loops, applies the following constraint:

$$\sum_{0 \le t \le T-1} v_{t \to t+1} = 0 \implies \sum_{0 \le t \le T-1} e(v_{t \to t+1}) = 0.$$

We wish to show that this constraint implies that $e$ must be a linear function over the reals. To show this, we first prove homogeneity, i.e., $e(\alpha v) = \alpha e(v)$ for all $\alpha \in \mathbb{R}$. Then we will prove additivity, i.e., $e(v_1 + v_2) = e(v_1) + e(v_2)$.

**1. Homogeneity:**

Consider a trajectory where $T = 2$, such that $v_{0 \to 1} = -v_{1 \to 2} = v$, and hence $v_{0 \to 1} + v_{1 \to 2} = 0$. Loop-closure implies:

$$e(v) + e(-v) = 0 \tag{1}$$
$$\implies -e(v) = e(-v). \tag{2}$$

This shows that $e$ is an odd function, so $e(0) = 0$.

Now, consider a trajectory where $T = n + 1$ for any $n \in \mathbb{N}$:

$$v_{t \to t+1} = \begin{cases} v & \text{for } 0 \le t \le n - 1, \\ -nv & \text{for } t = n. \end{cases}$$

Loop-closure implies:

$$ne(v) + e(-nv) = 0 \implies -ne(v) = e(-nv).$$

Using Eq. 2, we thus obtain

$$e(nv) = ne(v), \quad \forall n \in \mathbb{Z} \tag{3}$$

Thus, $e$ is homogeneous over the integers.

To show that $e$ is homogenous over the rationals, consider the loop with $T = 2$ given by $v_{0 \to 1} = mv$ for $m \in \mathbb{Z}$, and $v_{1 \to 2} = -nw$ for $n \in \mathbb{Z}^+$ with $w = \frac{mv}{n}$. The loop-closure condition implies to:

$$e(mv) + e(-nw) = 0 \tag{4}$$
$$\implies me(v) \quad = ne\left(\frac{m}{n}v\right) \tag{5}$$
$$\implies \frac{m}{n}e(v) \quad = e\left(\frac{m}{n}v\right). \tag{6}$$

Thus $e(\alpha v) = \alpha e(v)$ for $\alpha \in \mathbb{Q}$. Assuming that $e$ is a continuous function, and since the rationals are dense in the reals, $e(\alpha v) = \alpha e(v)$ for $\alpha \in \mathbb{R}$, i.e., $e$ is homogeneous over the reals.

**2. Additivity:**

Assume $T = 3$ with $v_{0 \to 1} = v$, $v_{1 \to 2} = w$, and $v_{2 \to 3} = -(v + w)$. Loop-closure implies:

$$e(v) + e(w) + e(-(v + w)) = 0 \tag{7}$$
$$\implies e(v) + e(w) \quad = -e(-(v + w)) \tag{8}$$
$$\implies e(v) + e(w) \quad = e(v + w), \tag{9}$$

for all velocity vectors $v$ and $w$.

**Conclusion:** Since $e$ is both homogeneous and additive, $e$ is linear over the reals. Thus, the loop-closure loss constrains the estimated velocities to be a linear function of the true velocities, thereby inducing a global metric structure in this estimated space.

## B.2 Data Ablations

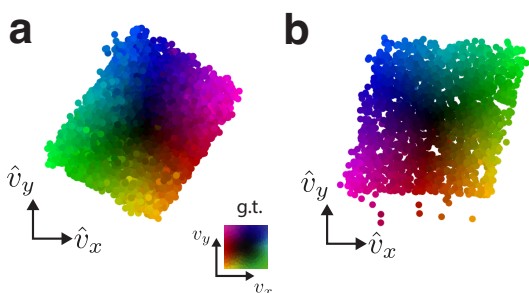

Figure 12: **Data Ablation studies. a.** Data ablation studies in the 2D Stretchy Bird environment show that training data does not need to consist only of loops. We created a dataset with 50% of trajectories as loops and 50% as independent random walks. The loop-closure loss applies only on loops while other loss terms apply on all trajectories. We report a transformation error of $0.035$, comparable to models trained on only closed-loop trajectories. **b.** The training paradigm we previously employed asked the model to predict $\hat{i}_{t+2}$ given $\hat{v}_{t \to t+1}$ and $i_{t+1}$, assuming that $v_{t+1 \to t+2} = v_{t \to t+1}$. This was done by construction to ensure that image features of $i_{t+2}$ were not memorized in the velocity latent space. This paradigm is not necessary. We retrain our network on the 2D Moving Blobs task with trajectories that vary in velocity randomly at each time-step. We predict an estimated $\hat{v}_{t \to t+1}$ from $i_t$ and $i_{t+1}$, and then predict $\hat{i}_{t+1}$ from $\hat{v}_{t \to t+1}$ and $i_t$. We report a transformation error of $0.02$, comparable to models trained under our earlier training paradigm.

We conduct two ablation studies on our data generation process.

First, we note that while our models are trained with data consisting of closed-loop trajectories, this training paradigm is for convenience and not a necessity of our framework. If arbitrary random walks were selected instead for training samples, self-intersections would automatically lead to loops within subsequences of the walks. These loops could then be used for the loop-closure loss, with all other losses applied on all loop and non-loop trajectories. As a simple proof of concept, we generated a dataset of 2D Stretchy Bird trajectories with 50% of the trajectories as loops and the remainder as independent random walks. While the next-state prediction, isotropy, and shortcut losses applied to all trajectories, the loop-closure loss only applied to trajectories that formed loops. Training on this modified dataset achieves a transformation error of $0.035$ (cf. Fig. 12a), comparable to the error reported in Table 1 (corresponding to models trained on only closed-loop trajectories). In greater generality, we note that the loop closure loss can also work on trajectories that are "almost loops" – i.e., a loss whose coefficient scales with how close a trajectory forms a closed loop.

Second, to ensure that our model velocity latent space did not memorize image features, we estimate a velocity $v_{t \to t+1}$ from $i_t$ to $i_{t+1}$, and predict an unseen $i_{t+2}$ that is $v_{t \to t+2} = v_{t \to t+1}$ away from $i_{t+1}$. Fig. 12b shows that this training paradigm is not a necessity. We train on trajectories from the Moving Blob environment whose velocities vary at each timestep. Training our model to estimate a velocity $v_{t \to t+1}$ from $i_t$ to $i_{t+1}$ and predicting $i_{t+2}$ from $v_{t \to t+1}$ from $i_t$ also results in similar performance. We obtain a transformation error of $0.02$, again comparable to the error reported in Table 1 (corresponding to models trained under our earlier paradigm).

## C  Experimental Details

**Code** All experiments were run on a single NVIDIA Titan RTX GPU. Each experiment took anywhere from 1-5 hours to train. Code and all experimental runs can be found here: `https://github.com/abhi-iyer/velocity_extraction`.

**Dataset construction.** Details about our dataset construction can be found in Table 2. Each sample in the dataset is described as $\mathbf{x} \in \mathbb{R}^{T \times \text{frame size}}$, which is a state consisting of two parts where $T$ is the trajectory length:

- $\mathbf{x}_1 \in \mathbb{R}^{\frac{T}{2} \times \text{frame size}}$, is a random walk.
- $\mathbf{x}_2$, which is a negative permutation of $\mathbf{x}_1$ such that the velocities between states satisfy $\int_0^T v(t)\,dt = 0$.

**Loss function details.** The loss terms, whose specific coefficients are also described in Table 2, are explicitly written out here:

- **Next-state prediction loss.** Given two states $i_t$ and $i_{t+1}$, the next-state prediction loss minimizes the distance between the predicted state and the true state: $\min ||i_{t+2} - \hat{i}_{t+2}||_2$.
- **Loop-Closure Loss.** The loop-closure loss ensures that all the predicted velocities in a given trajectory sum to zero given that the trajectory is a loop: $\min \sum_{0 \leq t \leq T-1} \hat{v}_{t \to t+1} = \min || \oint \hat{v} dt ||$.
- **Shortcut Loss.** The shortcut loss ensures that the decoder $g$ can generalize given a state and a velocity. For instance, $\min ||g(i_{t+2}, \hat{v}_{2 \to 3} + \hat{v}_{3 \to 4}) - \hat{i}_{t+4}||$.
- **Isotropy loss.** Finally, the isotropy loss induces an isotropy in the inferred velocity space: $\min \text{var} \left[ ||\hat{v}_{t \to t+1}|| \mid d(i_t, i_{t+1}) < \theta \right]$, where $d$ is a similarity function in the input image space and $\theta$ is some small threshold.

**Best-fit linear transform as an error metric.** We aim for the estimated velocities $\hat{\mathcal{V}}$ to be a linear function of the true velocities $\mathcal{V}$. Correspondingly, we first estimate the best-fit linear transformation $T$ from $\hat{\mathcal{V}}$ to $\mathcal{V}$ via a pseudoinverse, $T = \mathcal{V}\hat{\mathcal{V}}^\dagger$. Then, we compute our error metric as a normalized mean-squared error between the transformed points and the true distribution: $e = \frac{||T\hat{\mathcal{V}} - \mathcal{V}||_2}{N \text{var}\{\mathcal{V}\}}$. The presence of outliers, particularly in some of the poorly performing baseline methods, can lead to particularly poor best-fits $T$. To alleviate this, we find the transformation $T$ after removing a small number of outliers in $\hat{\mathcal{V}}$ via the DBSCAN clustering algorithm (any outlier rejection tool will suffice); however, we report the error $e$ evaluated on the entire dataset including outliers.

**Grid cell model.** To visualize grid firing fields given inputs from our model, we use an approximation of a continuous attractor model for a module of grid cells: we simulate patterned activity on a lattice of neurons as the sum of three plane waves, resulting in a hexagonal pattern of activity. Input velocities are used to update the state of the activity on the lattice of neurons through updating the phases of the plane waves, leading to accurate integration of the input velocities.

| Hyperparams | Stretchy Blob (2D) | Stretchy Bird (2D) | Stretchy Bird (3D) | Moving Blobs (2D) | Frequency Modulation (1D) |
|---|---|---|---|---|---|
| Frame size | $16 \times 16$ | $32 \times 12$ | $32 \times 12$ | $16 \times 16$ | $1 \times 100$ |
| Trajectory length | 81 | 81 | 81 | 81 | 81 |
| Velocity distribution | $U^2(0.05, 0.6)$ | $U^2(-1.5, 1.5)$ | $U^3(-1.5, 1.5)$ | $U^2(-20, 20)$ | $U(0.1, 10)$ |
| Max velocity step | 0.08 | 1.5 | 1.5 | 1.0 | 0.05 |
| Optimizer | Adam | Adam | Adam | Adam | Adam |
| Learning Rate | 5e-4 | 5e-4 | 5e-4 | 5e-4 | 5e-4 |
| Epochs Trained | 800 | 800 | 800 | 800 | 1200 |
| Batch size | 256 | 192 | 192 | 256 | 256 |
| Learnable Parameters | 536e3 | 622e3 | 622e3 | 536e3 | 544e3 |
| State Prediction Weight | 1 | 1e1 | 1e1 | 1 | 1 |
| Loop-Closure Weight | 1e1 | 1e2 | 1e2 | 1e1 | 1e1 |
| Shortcut Estimation Weight | 1 | 1e1 | 1e1 | 1 | 1 |
| Isotropy Weight | 1e2 | 1e2 | 1e2 | 1e2 | 1e2 |
| Isotropy Threshold | 1e-4 | 6e-3 | 6e-3 | 1e-2 | 1e-4 |
| Training Set Size | 800e3 | 800e3 | 800e3 | 800e3 | 800e3 |
| Testing Set Size | 200e3 | 200e3 | 200e3 | 200e3 | 200e3 |

Table 2: Hyperparameters used for generating the datasets and training the networks.

