# OpenReview forum: "Flexible mapping of abstract domains by grid cells via self-supervised extraction and projection of generalized velocity signals"
_NeurIPS.cc/2024/Conference — NeurIPS 2024 poster_

### Official Review · Reviewer_AXig · 2024-06-23

**Soundness:** 3
**Presentation:** 3
**Contribution:** 2
**Rating:** 4
**Confidence:** 4

**Summary:**

This work concerns stimuli that come from an underlying low-dimensional latent space. It studies a path-integrating problem on this space, in which a network is shown two images, has to infer a displacement signal between these images, and use it to traverse the space in a few different ways specified by a series of losses. It uses this as a model of velocity extraction for abstract cognitive maps in neuroscience.

**Strengths:**

I agree with the choice of problem. The brain must be extracting abstract velocity signals when it is traversing structured spaces. This is therefore an interesting and relevant problem.

Further, the stated network and losses do, indeed, nicely solve the problem.

Finally, the exposition is, for the most part, clear and compelling. The figures were very nice.

**Weaknesses:**

First, I think the framing with respect to previous work is needlessly confrontational, leading to some silly claims. Currently it is framed as if the proposed approach is a fundamentally different way of constructing a cognitive map, I think this is wrong, rather it seems like a useful addition that could coherently be included in existing models, making them more powerful. Relative to CSCG, TEM and models like them, I think this work is simply answering a different part of the cognitive mapping question. As the authors correctly point out, these models require a velocity signal (though that signal has no semantics). They then learn the meaning of the velocity signal (for example that north is the opposite of south) and use that to create a representation that is claimed to match neural data. In contrast, the author's model is a model of velocity extraction, of turning pairs of points into a velocity signal. You could well imagine a combined model that extracted velocities according to the author's scheme, and used them to drive a CSCG/TEM like model (i.e. the function $g(i_1, v)$ would be CSCG/TEM). This would ideally have the cross-environment generalisation of CSCG/TEM (which the author's model doesn't have), and the cross-velocity-modality generalisation of the current model. I think SR is not really a model of cognitive mapping, so I'm not too worried about the comparisons there.

Related to this, the claim that the velocity signals are learned independently of the stimuli also seems misleading (figure 1g). The same network, f, cannot transfer across sets of stimuli, even if they share the same underlying structure, unlike CSCG/TEM. Have I misinterpreted, should I take this to mean the same velocity is extracted independent of where in the space that velocity begins?

The silly claims that I think this leads to are the claims of novel predictions. Everyone and their mother would have predicted, from the moment the stretchy birds experiment worked, that the cellular basis of the grid signal would be modules of grid cells behaving in the same way they do in physical space. Yet, in lines 78-82 and 257-265 the authors make exactly this series of claims and frame it as a novel contribution. This is certainly a prediction, but it is far from unique to this work! Every model of grid cells, of which there are likely hundreds, is ambiguous to whether the variable being encoded is physical or abstract space!

A further confusion for me was the requirement for the latent code to be a linear function of the true velocities. Why is it so important? (As is implicitly assumed in the loss calculated in the table, though I couldn't find the loss described in detail anywhere) I see that the shortcut estimation loss and the isotropy loss build this linearity assumption into the velocities (though it seems from the ablation studies that it is actually not necessary to make the representation mostly linear), but I wonder whether it is not more impressive the fewer losses you use? It adds a lot of precision and clarity to be able to say that just the next-state prediction loss is enough to get a nonlinear encoding of the velocity, and from the ablation study, then adding loop closure gets you a linear representation.

Finally, I thought both the comparison to other techniques, and the grid cell results were slightly gimmicky. That's fine, but I don't give them a huge amount of weight as a result. Of course, if you have a perfect velocity code and feed it to a hexagonal grid cell model it will look like a hexagon. Further, why should I expect other dimensionality reduction techniques to learn the velocity? It seems like that is an assumption about the dataset (that all pairs of points separated by the same velocity signal correspond to the most meaningful axes on which to describe differences between). PCA extracts a different, but also meaningful encoding. It is interesting that this is not the same subspace, and certainly it shows the proposed method is the best at this, but the phrasing made it sound like a key contribution is beating these other methods (lines 74-77). In reality, it seems like your loss is looking for a specific type of dimensionality reduction that your model is built for and the others aren't. Its still a non-trivial result, and don't get me wrong, using this technique for dimensionality reduction sounds interesting, but I think the framing is currently way off.

Two smaller points:

The stated equivalent between velocities summing to 0 and inputs being the same assumes no aliasing (i.e. no repeated stimuli at different positions) (bottom fig 2c), in reality the two are not equivalent (velocities summing to 0 implies $i_1 = i_N$, but not the other way around). It should either be relaxed to a one directional implication, or the assumption of no aliasing should be stated.

I think the velocities in the latent space were literally added (like vector addition) to compute the shortcut loss, is that true? If so, it should be highlighted that this limits the spaces in which this can be used to ones in which velocities commute. For example, unless I've misunderstood, you cannot then use this for spheres, this should be mentioned as a limitation.

**Questions:**

My questions are largely included in the weaknesses section. Further more details of the exact form of the losses would be nice.

**Limitations:**

Limitations were not very clearly discussed.

---

> ### Author Rebuttal · Authors · 2024-08-07
>
> We would like to thank the reviewer for their detailed feedback and for raising excellent questions. We will try to address all questions one-by-one.
>
>
> > **Relative to CSCG, TEM and models like them, I think this work is simply answering a different part of the cognitive mapping question**
>
> Thank you for highlighting these points and for your careful review!
> As we note in the text, the crucial difference between our approach and those of others such as TEM and CSCG, is the following: previous approaches perform simultaneous learning of representations of external states and transitions between states; in contrast, we propose that only transitions need to be learned, which are then processed by a reusable grid attractor to enable integration and mapping. Thus, velocity-extraction can be done more simply relative to models that would involve both representational embedding and transition learning. However, we completely agree with the reviewer’s point that our model can augment models such as TEM and CSCG, by taking advantage of our model’s state-independent extracted velocities (or equivalently, actions). We will note this in our revised text and we will correspondingly edit the framing of our comparison with background work, which we did not intend to be confrontational.
>
>
> > **Claim that the velocity signals are learned independently of the stimuli also seems misleading (figure 1g)**
>
> We claim that the extracted velocity signals are independent of the stimuli _within each environment_. We do not make any claims that a network trained on one environment should be out-of-the-box generalizable to other abstract domains. Fig. 1g shows the extraction of generalized velocity signals within an example environment, Stretchy Blob. What we are saying is that the same velocity is extracted at various points within a space, independent of the specific location in that space. We will clarify this additionally in the text.
>
>
> > **Yet, in lines 78-82 and 257-265 the authors make exactly this series of claims and frame it as a novel contribution. This is certainly a prediction, but it is far from unique to this work!**
>
> Thank you for the opportunity to clarify!
>
> Our prediction here is not simply that grid cells can encode physical and abstract space, or that grid cell-like tuning exists across different spaces. Our prediction is, rather, that cell-cell correlation is preserved _across_ different spaces. The key nuance in our framework is the use of a single prestructured grid cell attractor network across different spatial and non-spatial environments through being able to extract generalized velocities in each task domain. Because we hypothesize that the brain generates velocity signals and pipes them into a rigid, prestructured set of attractor networks, we can predict that the cell-cell correlations within each grid network should be preserved across different modalities (i.e. if two grid cells are co-active for a spatial task, they should remain coactive for a non-spatial task). Since other models (such as TEM) do not use prestructured attractor networks, and learn the appropriate representations from scratch in different domains, it is unclear whether the cell-cell correlations will be preserved across domains/modalities. Although our conclusions may seem straightforward, other computational models that build cognitive maps (such as TEM, CSCG, SR) do not use fixed, prestructured attractors. We are unaware of these models being able to make such a prediction.
>
>
> > **The requirement for the latent code to be a linear function of the true velocities**
>
> Since grid cells are doing path integration through linear addition of velocity vectors, constraining our latent space to also be a linear function of the true velocity signals was crucial. Indeed, as the reviewer correctly noted, linearity is built into our choice of losses. In fact, the loop-closure loss imposes a linearity constraint (as can also be seen through our ablation studies; we will add a brief explanation of why loop closure ensures linearity in the appendix), with the additional shortcut and isotropy losses refining the obtained linear representations of velocity estimates. Indeed, what the reviewer noted is accurate: next-state prediction leads to a nonlinear representations of velocities, loop closure makes the encoding linear, and additional terms then refine the representations. We thank the reviewer for this succinct description and will include it in the text.
>
>
> > **I couldn't find the loss [calculated in the table] described in detail anywhere**
>
> We apologize for not including these details. We will include additional details in the text describing calculation of the error used in Table 1. We provide a brief description of the error metric here.
> As discussed above, we aim for the estimated velocities $\hat{\mathcal{V}}$ to be a linear function of the true velocities $\mathcal{V}$. Correspondingly, we first estimate the best-fit linear transformation $T$ from $\hat{\mathcal{V}}$ to $\mathcal{V}$ via a pseudoinverse, $T = \mathcal{V} \hat{\mathcal{V}}^\dagger$. Then, we compute our error metric as a normalized mean-squared error between the $N$ transformed points and the true distribution: $e = \frac{|| T\hat{\mathcal{V}} - \mathcal{V} ||_2}{N \text{var}( \mathcal{V} ) } $. The presence of outliers, particularly in some of the poorly performing baseline methods, can lead to particularly poor best-fits $T$. To alleviate this, we find the transformation $T$ after removing a small number of outliers in $\hat{\mathcal{V}}$ via the DBSCAN clustering algorithm (any outlier rejection tool will suffice); however, we report the error $e$ evaluated on the entire dataset including outliers.

---

> ### Author Response · Authors · 2024-08-07
> **Rebuttal, Part 2**
>
> > **Why should I expect other dimensionality reduction techniques to learn the velocity?**
>
> > **PCA extracts a different, but also meaningful encoding. It is interesting that this is not the same subspace, and certainly it shows the proposed method is the best at this, but the phrasing made it sound like a key contribution is beating these other methods**
>
>
> This is an excellent question.
> By construction, the datasets that we have generated have a specified intrinsic dimensionality (1, 2 or 3, depending on the environment). For example, in the Stretchy Bird environment, there exists a minimal representation of the dataset that is captured by a two-dimensional coordinate, neck and leg lengths. Thus, within the high-dimensional space in which the raw inputs live, the dataset is explicitly constructed to lie on a (nonlinear) two-dimensional manifold. If a dimensionality reduction method were able to flatten this manifold and represent it in two dimensions, differences between points would produce a veridical representation of velocities.
> While PCA does provide an encoding of the data (that describes the projection of the data onto the principal components), it does not provide a meaningful representation, since it is unable to find even a low-dimensional representation of the data. We demonstrate our viewpoint further through the new Fig. R3 in the attached PDF. Here, we examine a dataset generated through a single trajectory within our 2D Moving Blobs environment. Through integration of our velocity estimates, we find that the high-dimensional states collapse onto a two-dimensional plane, and are thus completely captured by two coordinates. In contrast, PCA appears to require 24 dimensions to capture 95% of the variance within the data, and appears to occupy a volume when plotted in three dimensions. Since PCA is unable to find a low-dimensional representation of a dataset that is intrinsically two-dimensional, we believe that the projection learned by PCA is not strongly meaningful for this dataset. These new results continue to point towards dimensionality reduction as a key contribution and strength of our model. As a result, we felt that it was reasonable to use the other dimensionality reduction techniques as necessary baselines, which the other reviewers have appreciated. In summary, we find that integrating estimates of low-dimensional transitions between high-dimensional states can be an effective tool for dimensionality reduction.
>
>
>
>
> > **Stated equivalent between velocities summing to 0 and inputs being the same assumes no aliasing**
>
> Thank you for pointing this out. We had implicitly assumed that within the abstract cognitive spaces we investigate, states are unique at each point in space. We will make this assumption explicit in the text.
>
>
> > **You cannot then use this for spheres**
>
> Thank you for raising this subtlety in our model. We do in fact assume that the velocity vectors in the space of our latents commute. As a result, the estimated velocity vectors cannot directly represent tangent vectors in a non-Euclidean space, such as a sphere. We will add this note to the paper. However, this does _not_ entirely preclude the representation of these non-Euclidean spaces through our method, since we can consider these spaces as embedded in a higher dimensional Euclidean space wherein velocity vectors will commute again.
> For example, in representing a point in an abstract domain with spherical geometry, our method will fail if we attempt to estimate two-dimensional velocity vectors representing transitions. However, if instead we estimate three-dimensional vectors, we expect our method to continue to work since the surface of a sphere can be embedded in three-dimensional space. For data arising from a general manifold, we will require a few extra dimensions (cf. Whitney embedding and Nash embedding theorems) but will always be able to consider a Euclidean space of sufficient dimensions that will embed the data generating manifolds. We thank the reviewer for leading us to consider this subtlety and will include this cost of requiring extra dimensions for non-flat spaces as a limitation in the discussion section of our paper.

---

> ### Author Response · Authors · 2024-08-08
> **Rebuttal, Part 3**
>
> > **Further more details of the exact form of the losses would be nice**
>
> Thank you for the comment. We have currently listed the description of each loss in L173-195 along with equations in Fig. 3 and are happy to explicitly write out our loss functions and other algorithmic details in the appendix, that we briefly mention here:
>
> Given two states $i\_t$ and $i\_{t+1}$, the next-state prediction loss minimizes the distance between the predicted state and the true state: $\text{min} ~ || i\_{t+2} - \hat{i}\_{t+2}||\_2$.
>
> The loop closure loss ensures that all the predicted velocities in a given trajectory sum to zero given that the trajectory is a loop: $\text{min}  \sum\_{0 \leq t \leq T-1} \hat{v}\_{t \rightarrow t+1} = \text{min} ~ || \oint \hat{v} dt ||$.
>
> The shortcut loss ensures that the decoder $g$ can generalize given a state and a velocity. For instance, $\text{min} ~~ || g(i\_{t+2}, \hat{v}\_{2 \rightarrow 3} + \hat{v}\_{3 \rightarrow 4}) - \hat{i}\_{t+4} ||$.
>
> Finally, the isotropy loss induces an isotropy in the inferred velocity space: $\text{min} ~ \text{var} [ ||\hat{v}\_{t \rightarrow t+1} || ~ \mid ~ d(i\_t, i\_{t+1}) < \theta ]$, where $d$ is a similarity function in the input image space and $\theta$ is some small threshold.
>
> All losses have a weight / prefactor, listed in Table 2 of our paper. As mentioned in L200, regardless of the training environment, the relative weighting of the two critical loss terms remains consistent (the loop-closure loss weighted ten times higher than the next state prediction loss). We also plan to discuss more about how each synthetic task was generated, as also recommended by $\textbf{sK54}$.

---

> ### Author Response · Authors · 2024-08-11
>
> We hope that you agree that our paper has improved given your detailed feedback and suggestions! Given that we have addressed the primary concerns raised in the review, we kindly ask you to adjust your score while taking our rebuttal into account.

---

> > ### Comment · Reviewer_AXig · 2024-08-11
> > **Response**
> >
> > Thank you for your extensive response, I appreciate the assiduousness and engagement.
> >
> > Broadly, the comments answer my questions, especially about aliasing and spheres.
> >
> > I continue to disagree about the importance of the dimensionality reduction comparison to e.g. PCA. Sure, you generate datasets in which differences are a better way to extract out the underlying space than PCA. Your method, aimed at extracting velocities, therefore does better. That is wonderful, but I believe my original comments still hold.
> >
> > Lastly, you got me musing about what it is we do when we build such models of the brain (cue authors' groan). I continue to think that the prediction of preserved cell-cell correlations across domains is what everyone and their mother (I checked with my mother) would have thought post-stretchy birds experiment, perhaps I am too new to the field and missed some 00/10s era debate on this. This model shows that, if you can design a scheme for extracting velocities, such an idea works.
> >
> > But, could it ever not have done? If you are capable of extracting velocities from pairs of images correctly then of course you can pipe them into a reusable grid cell system in which such cell-cell correlations are preserved. Did we need a model to tell us that?
> >
> > The novelty of your work seems to be in the design of a system to extract velocities, the losses etc. that can make it work, and using it to do dimensionality reduction on appropriate datasets. Hypothetically, you could make the same attack at models like TEM/CSCG/SR, that they take a computation, and simply show a model of it working, but (a) I think I learnt more from the algorithmic details needed than I did here and much more importantly (b) they predict neural behaviour!
> >
> > So it seems the potential neuroscience value I am drawing from this model is about the losses that are required to make it work. I do find it interesting so many losses are needed to make it perform well, I would have guessed that just next-state prediction and your architectural choices might have done it (and who knows, with another setup it still might). But I'm afraid given my above thoughts, despite the paper being well-explained, thorough, correct, and generally nice science, I will keep my score.
> >
> > If you think there is something that has still not made it through my thick skull please do let me know.

---

> > > ### Author Response · Authors · 2024-08-12
> > > **Response, Part I**
> > >
> > > Thank you for your response to our rebuttal, and for recognizing our work as being well-explained, thorough, correct and containing generally nice science.
> > >
> > >
> > > > **I continue to disagree about the importance of the dimensionality reduction comparison to e.g. PCA. Sure, you generate datasets in which differences are a better way to extract out the underlying space than PCA. Your method, aimed at extracting velocities, therefore does better. That is wonderful, but I believe my original comments still hold.**
> > >
> > > We do not believe our improved performance in comparison with PCA is simply because of our choice of datasets. Through the idea of velocity extraction as a means to perform dimensionality reduction, we can leverage local transition structures in the original data manifold that sequence agnostic methods (e.g., PCA, UMAP, etc.) cannot use. Thus, our method can utilize the crucial trajectory information in the data to perform more efficient dimensionality reduction. In this context, we do acknowledge that our method assumes the presence of trajectories characterized by continuous variation in inputs; however, we believe this is not a restrictive assumption, as many datasets, including video data, typically fall into this category. An examination of larger, more realistic datasets lies within the scope of future work that we plan to pursue.
> > >
> > > You mention that you believe that your original comments in this context still hold. Our understanding of your original comments were that you had three primary questions/concerns: (1) One may not expect other dimensionality reduction methods to extract a velocity. (2) PCA may still be extracting a useful embedding that happens to not correspond to the velocities that we extract. (3) Our loss terms specifically looked for a specific type of dimensionality reduction that other methods did not.
> > > In our earlier response, we attempted to respond to each of these concerns: (1) We argued that any dimensionality reduction method should be expected to extract a velocity, since the data lies on a very low-dimensional manifold embedded in the high-dimensional space. (2) Since PCA was unable to find a low-dimensional representation, we argued that it was not a meaningful dimensionality reduction. (3) _Yes_, our loss terms do look for dimensionality reduction based on transitions which other methods are unable to take into account; this allows us to be able to perform more effective dimensionality reduction as evidenced by extraction of a two-dimensional embedding of the data in a dataset where other methods failed to extract low-dimensional representations, even though the original data lived on a two-dimensional manifold.
> > >
> > > We hope this response has helped to clarify the salience of our approach and address the concerns you raised in your original comments. If there are any unresolved questions about the importance of our method as a dimensionality reduction technique, we would appreciate further discussion to understand your perspective.

---

> > > ### Author Response · Authors · 2024-08-12
> > > **Response, Part II**
> > >
> > > > **Prediction of preserved cell-cell correlations across domains is what everyone and their mother (I checked with my mother) would have thought post-stretchy birds experiment…But could it ever have not been done? If you are capable of extracting velocities from pairs of images correctly then of course you can pipe them into a reusable grid cell system in which such cell-cell correlations are preserved. Did we need a model to tell us that?**
> > >
> > > Prediction of cell-cell correlations across domains of different modalities may appear as a straight-forward prediction, however, it critically depends on two key ideas: (1) that low-dimensional velocities can indeed be extracted from abstract domains, and (2) that the _same_ continuous-attractor-based grid module can perform integration across domains. The bulk of our work establishes (1), through constructing an SSL framework for this velocity extraction that performs better than (both deep and non-deep) baselines.
> > >
> > > Note that simply the assumption of a single continuous-attractor-based grid module would not be sufficient, since typical continuous attractor models require a low-dimensional velocity input, and one would thus need to first establish that velocities can be extracted across inputs from different modalities. To summarize, in making predictions for grid cell correlations, we assume a continuous attractor network based grid model and provide a computational basis for linking such an attractor to inputs from different modalities through our framework for velocity extraction.
> > >
> > > To establish that across domain cell-cell preservation is a nontrivial prediction, and do not follow directly from pre-existing literature, we point towards the other related models of cognitive mapping mentioned by the reviewer: (1) TEM learns grid cell representations from scratch in any given environment. If presented with a new environment, TEM requires re-learning of the affordances and statistics of the new environment, and as a result cells that were correlated within an environment of one modality are not guaranteed to be correlated in a different modality. (2) SR does not produce preserved cell-cell correlation structure across environments within the same modality itself, and (3) CSCG does not provide grid responses to be able to compare cell-cell correlations.
> > >
> > > In the context of these above models, we hope it is clearer that cell-cell correlations being preserved across modalities does not trivially follow from previous modeling work. We are also unaware of such a conclusion being drawn in the literature in the light of previous experimental work, such as the stretchy bird experiments, sound modulation experiments, or any others. We would be grateful if the reviewer could point us towards such a reference that has already made the predictions that we have stated in our work.
> > >
> > > > **I think I learnt more from the algorithmic details needed than I did here and much more importantly (b) they predict neural behaviour!**
> > >
> > > The crucial difference between our approach and those of others such as TEM and CSCG, is the following: previous approaches perform simultaneous learning of representations of external states, and transitions between states; in contrast, we propose that only transitions need to be learned, which are then processed by a reusable grid attractor to enable integration and mapping.
> > > Approaching the cognitive mapping problem from a velocity-extraction-first perspective, means that the mapping of the environment to grid states can be performed simply by a prestructured continuous attractor network, instead of something more complex like TEM. Thus, in terms of mapping spaces, our model can be thought of as making predictions for neural behavior corresponding to those obtained from the continuous attractor model, which can be distinct from the predictions made by the grid cells in other hippocampal models. If one wishes to examine neural predictions for hippocampal cells rather than grid cells, our method points towards the usage of a hippocampal complex model that can use prestructured attractor dynamics, such as Chandra et al. 2023 bioRxiv 2023.11.28.568960v2.

---

> > > ### Author Response · Authors · 2024-08-12
> > > **Response, Part III**
> > >
> > > > **I do find it interesting so many losses are needed to make it perform well, I would have guessed that just next-state prediction and your architectural choices might have done it**
> > >
> > > The number of losses needed for our model to “perform well” effectively depends on what criteria are being imposed in defining “well”.
> > >
> > > If we are interested in extracting some nonlinear low-dimensional velocity between states in any abstract domain, then just the next-step prediction loss is sufficient, as seen in Fig. 9. If we additionally demand that the estimated velocities be linear functions of the ground truth velocities, then the loop-closure loss needs to be added. If we then additionally demand isotropic representations of velocities, we need to include our isotropy loss. And, if we require further refinement, with very precise encoding of velocities, we include our shortcut loss. As indicated in our earlier rebuttal, we will certainly include an updated description of our model results to clarify this point.
> > >
> > > As such, we argued earlier that grid cells would require linear representations of velocities and thus posited that next-step prediction and loop-closure were essential losses to consider. We included isotropy and shortcut losses as additional auxiliary losses to help refine the obtained solution, and will make it clearer in the text that these are not essential to any of our key results. We are happy to include an elaboration of this discussion in text, and will also include a proof to show that loop-closure loss exacts a strict linearity constraint on its inputs.
> > >
> > >
> > > > **But I'm afraid given my above thoughts, despite the paper being well-explained, thorough, correct, and generally nice science, I will keep my score.**
> > >
> > > We hope our responses have further established the key interesting aspects of our results and would appreciate a re-evaluation of our score in this context.

---

> > > > ### Comment · Reviewer_AXig · 2024-08-12
> > > > **Final Thoughts**
> > > >
> > > > I thank the authors for their further extended commentary. In summary, I'm afraid I still remain unconvinced on both counts, and will keep my score.
> > > >
> > > > On the PCA point: there are problems where velocity extraction is key, and your method does well. There are others where there is no dynamical structure, and top PCs do well. There are probably also those with dynamical structure but on which PCA does better. Different problems demand different algorithms. Nonetheless, it's cool that yours does well.
> > > >
> > > > On the preserved cell-cell correlation count: I realise that no other model predicts this. My point was more that I don't think we needed a model to tell us this would work. I agree probably no-one published this, but at the risk of sounding too harsh, I think that's likely because no-one thought they needed to - of course piping velocities into a pre-existing CAN would work like that? Further, there's a separation between a model of learning, and a model of flexible re-use of past learnt things, both CSCG and TEM seem like models of the learning of structure rather than statements that 'of course this structure never gets re-used'.
> > > >
> > > > Finally, the loss points you make are very interesting.

---

> > > > > ### Author Response · Authors · 2024-08-12
> > > > >
> > > > > Thank you for your comments. We summarize our discussion here for convenience, please let us know if you disagree. You raised the following key comments/criticisms of our work:
> > > > >
> > > > >
> > > > > 1. **Of course piping velocities into a pre-existing CAN would work.** This criticism is analogous to “if you solve the problem and get good estimates (i.e., extracting these geometrically consistent velocities), using the estimates for computation will work (i.e., pipe them into a CAN and integrate within the abstract domain)”. This criticism skips over the crucial prerequisite challenge of extracting these velocities in the first place (i.e., solving the problem), which we crucially demonstrate in our work.
> > > > > 2. **TEM-like models are potentially better because they predict neural behavior.** Yet, the key prediction that you suggest is trivial to achieve (correlations being preserved across domains) cannot be replicated in these models.
> > > > > 3. **Dimensionality reduction methods should not be able to extract velocity.** However, we showed that since the data lies on a low-dimensional manifold, a dimensionality reduction method should lead to velocities if the low-dimensional structure was identified. Our method identifies this structure and is thus able to extract a consistent representation of velocity.
> > > > > 4. **Our method works specifically on datasets with dynamics.** Yes, we agree and believe this is a strength of our work that sets it apart from other methods. The synthetic tasks we use make it highly relevant to big data applications today where timeseries and video data are increasingly gaining importance.
> > > > > 5. **We require too many losses.** We specifically demonstrated that this is not the case, and that one can add more losses depending on what qualities are desired from the velocity representations.
> > > > > 6. **We cannot work with non-Euclidean domains.** We show how our model can still work in nonlinear, non-flat manifolds as well.
> > > > >
> > > > >
> > > > > In summary, you conceded that our work is well-explained, is thorough, correct, and contains nice science. Further, you stated that the choice of problem was relevant and interesting, and that our exposition was clear and compelling. While we recognize that you appear to remain unconvinced by our arguments, we hope that this summarization can lead to a reconsideration of your position on our work.

---

### Official Review · Reviewer_SEuj · 2024-07-09

**Soundness:** 4
**Presentation:** 4
**Contribution:** 3
**Rating:** 8
**Confidence:** 4

**Summary:**

This paper develops a new dimensionality reduction approach based on velocity in latent space. It is inspired by, and tries to emulate, aspects of entorhinal grid cell activity. The critical ingredient is a "loop-closure" constraint that drives the model to build a metric map of the latent space. The results demonstrate advantages of this approach over traditional dimensionality reduction algorithms.

**Strengths:**

- The paper proposes a novel solution to an interesting and important problem.

- Writing and visualizations are very clear (some minor comments on this below).

- The experimental results are impressive relative to baselines.

- The work will be potentially impactful within neuroscience. I'm less sure about AI (see Weaknesses).

- I appreciated that the authors laid out the predictions of their framework. It would be great to see these carefully tested.

**Weaknesses:**

- With respect to the brain, I think the model makes some assumptions with questionable plausible (see Questions below), but I would be interested to hear if the authors disagree.

- I didn't feel that the authors made a really compelling case for AI applications, though I understand the potential utility in principle.

Minor:

- p. 3: Technically the state-based SR doesn't assume action inputs, since it's implicitly taken an expectation over actions. It's true that the state-action SR does take actions as input.

- p. 3: "learns simultaneously learns" -> "simultaneously learns"

- p. 8: "These inferred velocities" -> "With these inferred velocities"

- p. 8: "allow reuse grid cells" -> "allow reuse of grid cells"

- Somewhere in the supplement, the authors should completely spell out their loss function and other algorithmic details.

**Questions:**

- How does the model deal with boundary/obstacle effects? This seems like a situation where the effect of the velocity operator is not state-independent.

- Related to the previous point, if I've understood correctly, the model assumes that state variables live in R^D (it would be helpful if this was made explicit), so there are no boundary conditions. How do the authors think about this in relation to the toroidal topology that appears to constrain grid cell population activity?

- If the claim is that this model emulates how the entorhinal cortex learns velocity-based representations, I wonder how the authors think about the fact that often the training data animals receive will not have that many loops. For example, how many loops were there in the experiments on stretchy birds? What happens if the model doesn't have loops in its training set? What about trajectories that are *almost* loops?

**Limitations:**

The authors dicuss future directions at the end, but don't directly address limitations. There are no potentially negative societal impacts.

---

> ### Author Rebuttal · Authors · 2024-08-07
>
> We would like to thank the reviewer for their detailed feedback and for raising excellent questions. We will try to address all questions one-by-one.
>
> > **State-based SR doesn't assume action inputs**
>
> Thank you for this note. We will clarify this in our text and distinguish between state-based SRs and state-action SRs. We additionally note that the original SR paper (Dayan 1993) and recent extensions (Stachenfeld 2017) use a discrete action space whether SR is state-based (i.e. expectation over actions) or state-action based.
>
> > **Grammatical changes in text**
>
> Thank you for suggesting these revisions! We will change them in text.
>
> > **Somewhere in the supplement, the authors should completely spell out their loss function and other algorithmic details**
>
> Thank you for the comment. We have currently listed the description of each loss in L173-195 along with equations in Fig. 3 but are happy to explicitly write out our loss functions and other algorithmic details in the appendix, that we briefly mention here:
>
> Given two states $i\_t$ and $i\_{t+1}$, the next-state prediction loss minimizes the distance between the predicted state and the true state: $\text{min} || i\_{t+2} - \hat{i}\_{t+2}||\_2$.
>
> The loop closure loss ensures that all the predicted velocities in a given trajectory sum to zero given that the trajectory is a loop: $\text{min} \sum\_{0 \leq t \leq T-1} \hat{v}\_{t \rightarrow t+1} = \text{min} || \oint \hat{v} dt ||$.
>
> The shortcut loss ensures that the decoder $g$ can generalize given a state and a velocity. For instance, $\text{min} || g(i\_{t+2}, \hat{v}\_{2 \rightarrow 3} + \hat{v}\_{3 \rightarrow 4}) - \hat{i}\_{t+4} ||$.
>
> Finally, the isotropy loss induces an isotropy in the inferred velocity space: $\text{min} ~ \text{var} \left[ ||\hat{v}\_{t \rightarrow t+1} || \mid d(i\_t, i\_{t+1}) < \theta \right]$, where $d$ is a similarity function in the input image space and $\theta$ is some small threshold.
>
> All losses have a weight / prefactor, listed in Table 2 of our paper. As mentioned in L200, regardless of the training environment, the relative weighting of the two critical loss terms remains consistent (the loop-closure loss weighted ten times higher than the next state prediction loss). We also plan to discuss more about how each synthetic task was generated, as also recommended by $\textbf{sK54}$.
>
>
> > **Boundary/obstacle effects**
>
> This is an excellent question. We clarify that while the decoder is state dependent by construction (outputs a predicted state given a state and velocity), the encoder is state-independent since it infers a generalized notion of velocity between any two high-dimensional states within our abstract domains. We investigate how the decoder performs at boundaries (Fig. R4 in the attached PDF) within the 2D Stretchy Bird environment. We find that the decoder implicitly understands the boundaries of the training data’s underlying manifold. That is, once a velocity produces a bird state that is unseen in the training distribution, i.e. a bird that cannot further shrink its legs or extend its neck, further transitions in the same direction do not produce any changes in the predicted state. Thus, the decoder understands the boundaries observed in the training data.
>
> > **Model assumes that state variables live in R^D**
>
> Thank you for raising this point! We do indeed assume that the input state variables occupy a region of $\mathbb{R}^D$ (where $D$ is the number of pixels in the input, for example); we also further assume that the estimated velocities live in $\mathbb{R}^d$ for some smaller $d$. This in and of itself is not particularly constraining: while a single module of grid states live on a torus, they can integrate velocities that are obtained generally from $\mathbb{R}^2$ without bounds. While grid states wrap around, the velocities do not necessarily need to wrap around boundaries. Further, if we include additional grid modules, grid cells can integrate velocities in the higher dimensional Euclidean space $\mathbb{R}^d$, as has been shown by recent theoretical work (Klukas et al. 2020, PLOS Computational Biology 16(4): e1007796). We are happy to add a discussion of this point in text.
>
> > **Concerns on loops**
>
> Thank you for raising these excellent points!
> While our models were trained with data consisting of loops, this training paradigm is simply for convenience (L179-181) and is not a necessity of the model. If arbitrary random walks were selected instead for training samples, self-intersections would automatically lead to loops within sub-sequences of the walks (as detected by $k<m$ with $i_k=i_m$) — these loops could then be used for the loop-closure loss, with all other losses applied on all loop and non-loop trajectories. As a simple proof of concept, we generated a dataset of arbitrary trajectories with 50% of the trajectories as loops and the remainder as non-loops. We applied loop closure loss on trajectories that formed loops and all other losses on the entire dataset of trajectories. Fig. R1 in the attached PDF demonstrates this paradigm for the 2D Stretchy Bird environment, where we continue to get consistent velocity representations with low transformation error to the true velocity distribution.
>
> In greater generality, we also note that the loop closure loss can also work on trajectories that are “almost loops” – i.e., a loss whose prefactor term scales with how close a trajectory is to forming a closed loop. However, running this analysis and tuning these parameters is beyond the scope of the current rebuttal period, and we will examine these results for the camera ready version of our paper.
>
> > **Lack of limitations**
>
> We apologize for not including sufficient discussion of limitations. We will add a discussion of the limitations to our model in text, as also mentioned to Reviewer $\textbf{sK54}$.

---

> ### Author Response · Authors · 2024-08-07
> **Rebuttal, Part 2**
>
> > **Compelling case for AI applications**
>
> We believe our work has potential application to machine learning beyond neuroscience, primarily through the context of dimensionality reduction.
>
> A crucial step in analysis of large datasets of high-dimensional data is often the generation of dimensionality reduced representations of the data. Our method provides a novel SSL framework for dimensionality reduction and manifold learning, which appears to significantly outperform baselines on datasets that contain relatively lower-dimensional transitions. Video data, for example, naturally contains transitions between frames that are typically lower dimensional than the frames themselves, and thus may be particularly amenable to our techniques.
>
> Moreover, typical nonlinear dimensionality reduction techniques, such as Isomap and UMAP are non-invertible. In contrast, similar to an autoencoder, we can use the decoder in our model to generate the high-dimensional states that correspond to points within the low-dimensional space. In this sense our model also acts like a generative model, by being able to generate inputs that correspond to different regions of the low-dimensional space.
>
> Furthermore, our method naturally lends itself to manifold alignment, which is particularly effective when the data exhibits a small number of continuous modes of variability. Through mapping two spaces onto the same low-dimensional space of transitions, our technique can perform equivalent transformations on datasets from different modalities. Moreover, with a small number of “gluing” points, our method allows for building one-to-one correspondences between different domains.
>
> We will be glad to include these discussion points in the camera ready version of the text.

---

> > ### Comment · Reviewer_SEuj · 2024-08-07
> > **Good points**
> >
> > Thanks for clarifying this! I think these are reasonable points to add to the discussion.

---

> > > ### Author Response · Authors · 2024-08-11
> > >
> > > We hope that you agree that our paper has improved given your detailed feedback and suggestions! Given that we have addressed the primary concerns raised in the review, we kindly ask you to adjust your score while taking our rebuttal into account.

---

> > > > ### Comment · Reviewer_SEuj · 2024-08-11
> > > > **score adjustment**
> > > >
> > > > My score was already quite high (8). While I really like this paper, I don't think it makes sense to raise it to 9, which would imply that this is a groundbreaking paper. I feel that is a valuable contribution and should definitely be accepted (I will advocate for it), but it's not going to revolutionize either neuroscience or machine learning. I agree that there are some potentially powerful contributions to AI, but these were not demonstrated in this paper.

---

> > > > > ### Author Response · Authors · 2024-08-11
> > > > >
> > > > > Thank you for your comments and thank you for your appreciation / advocacy of our paper!

---

### Official Review · Reviewer_SMXS · 2024-07-12

**Soundness:** 3
**Presentation:** 2
**Contribution:** 3
**Rating:** 7
**Confidence:** 4

**Summary:**

This work proposes a novel neural circuit model that explains how grid cells in the medial entorhinal cortex can map both spatial and abstract cognitive spaces. The model extracts low-dimensional velocity signals from high-dimensional abstract domains using self-supervised learning, leveraging grid cells’ spatial velocity integration capabilities. This allows the creation of efficient cognitive maps across different domains.

**Strengths:**

1.	Although the idea of using grid cells as a scaffold for representing cognitive maps is not new, the ability to apply continuous actions (‘velocity’ in this paper) and learn from the data is indeed a highlight.
2.	The paper conducts comprehensive ablation studies to verify the rationality of its objective function.
3.	The comparison with baselines is also quite thorough.

**Weaknesses:**

1.  The paper’s starting point is that grid cells can serve as a scaffold for different cognitive domains, as reflected in the title, but the main content focuses on the extraction of velocity. There is little discussion about grid cells, and the description of how the continuous attractor network of grid cells works is insufficient.
2.	The paper compares the performance of various baselines. The experimental results show that the presented model significantly outperforms the baselines in dimensionality reduction, but there is no clear explanation for why this model is superior to the baselines.
3.	The training process relies on specially constructed datasets (loops). It’s obvious that organisms do not need to return to the original point to achieve good representations.
4.  The authors didn’t discuss the situation when the dimension of velocity is higher than grid cells (2D).

**Questions:**

Please refer to weaknesses.

---

> ### Author Rebuttal · Authors · 2024-08-07
>
> We would like to thank the reviewer for their detailed feedback and for raising excellent questions. We will try to address all questions one-by-one.
>
> > **There is little discussion about grid cells, and the description of how the continuous attractor network of grid cells works is insufficient.**
>
> Thank you for highlighting this. We briefly describe why extracting velocity signals from non-spatial, abstract domains is an important prerequisite challenge for grid cells to path integrate between states in these domains (L35-38, L95-98, and Fig. 1 caption). Here we provide an additional explanation: continuous attractor models for grid cells have gained much favor through repeated confirmation via experimental results. However, relating these continuous attractor models to mapping of abstract domains requires the extraction of a faithful representation of velocity in these domains. This is in contrast to other models for spatial mapping, such as TEM, which learn grid cell-like representations from scratch while mapping new abstract spaces. Our work shows how it can be sufficient to use a prestructured attractor network and the experimental predictions that can arise from this modeling choice. We will elaborate on this discussion and make the connection back to grid cells clearer in the text.
> We apologize for not including a sufficiently detailed description of the continuous attractor network that we used. We provide here a brief description of the grid cell model used and will include a more detailed description in the appendix of the revised paper.
> We use an approximation of a continuous attractor model for a module of grid cells: we simulate patterned activity on a lattice of neurons as the sum of three plane waves, resulting in a hexagonal pattern of activity.  Input velocities are used to update the state of the activity on the lattice of neurons through updating the phases of the plane waves, leading to accurate integration of the input velocities.
>
>
> >  **Why this model is superior to the baselines**
>
> Thank you for highlighting this point as well. While we have briefly discussed why our model outperforms other dimensionality reduction techniques in L102-105, we elaborate on this further here and will include this in the paper:
>
> Typical dimensionality reduction techniques are agnostic to the transition structure between high-dimensional input states when drawn from a trajectory. They usually rely on the statistics of distances between points across the entire ensemble of inputs. In contrast, our method explicitly looks for a structured tangent manifold around each point that captures the low-dimensional transitions between successive states along a trajectory (see Fig. R3). Through using this additional sequential information, with a prior of assuming the existence of a tangent manifold along the manifold of input points, we obtain a superior performance. We have also previously included comparison to another technique that uses sequential information (MCNet). However, there too we find that our method is significantly better through assuming the existence of a low-dimensional description of transitions in a state-independent fashion, rather than a high-dimensional maximal description of the state-dependent transitions.
>
> > **Training process relies on specially constructed datasets (loops)**
>
> Thank you for raising this excellent point!
> While our models were trained with data consisting of loops, this training paradigm is simply for convenience (L179-181) and is not a necessity of the model. If arbitrary random walks were selected instead for training samples, self-intersections would automatically lead to loops within sub-sequences of the walks (as detected by $k<m$ with $i_k=i_m$) — these loops could then be used for the loop-closure loss, with all other losses applied on all loop and non-loop trajectories. As a simple proof of concept, we generated a dataset of arbitrary trajectories with 50% of the trajectories as loops and the remainder as non-loops. We applied loop closure loss on trajectories that formed loops and all other losses on the entire dataset of trajectories. Fig. R1 in the attached PDF demonstrates this paradigm for the 2D Stretchy Bird environment, where we continue to get consistent velocity representations with low transformation error to the true velocity distribution.
>
> In greater generality, we also note that the loop closure loss can also work on trajectories that are “almost loops” – i.e., a loss whose prefactor term scales with how close a trajectory is to forming a closed loop. However, running this analysis and tuning these parameters is beyond the scope of the current rebuttal period, and we will examine these results for the camera-ready version of our paper.
>
> > **Situation when the dimension of velocity is higher than grid cells (2D)**
>
> Thank you for raising this point. We have included an example of an environment where the dimension of the velocity was higher than two-dimensional, cf. the 3D Stretchy Bird environment. This, however, does not pose a problem for integration by grid cells. While a single grid cell module can only integrate two-dimensional velocities, previous theoretical work (Klukas et al. 2020, PLOS Computational Biology 16(4): e1007796) has shown that multiple grid cell modules can collectively represent large volumes of high-dimensional Euclidean spaces through integration of velocities in higher dimensions. We will add a brief discussion of this point in the text.

---

> > ### Comment · Reviewer_SMXS · 2024-08-12
> >
> > Thank you for your comments. I believe my concerns are properly addressed and I would like to maintain my score.

---

> > > ### Author Response · Authors · 2024-08-12
> > >
> > > Thank you again for your detailed feedback and suggestions. We hope you agree that our paper has improved as a result of this discussion. Please let us know if there are any additional questions that we may be able to address to consider a re-evaluated score.

---

> ### Author Response · Authors · 2024-08-11
>
> We hope that you agree that our paper has improved given your detailed feedback and suggestions! Please let us know if there is anything else that we can clarify. We kindly ask you to consider revising your score if you agree that the primary concerns raised in the review were addressed.

---

### Official Review · Reviewer_sK54 · 2024-07-13

**Soundness:** 3
**Presentation:** 4
**Contribution:** 3
**Rating:** 7
**Confidence:** 4

**Summary:**

This work explores how the brain could theoretically generalize its representations of velocity signals used to map spatial domains into abstract velocity signals that map abstract cognitive domains.

To do so, proposes a self-supervised ANN algorithm and architecture to learn a low-dimensional latent space that describes abstract velocities (transformation parameters) between sensory inputs resulting from a parameterized low-dimensional generative process (where parameters are points in abstract space). For example, in a Stretchy Blob task, an image of a 2D Gaussian blob is the product of a generative process parameterized by[width, height], and the difference between two images/blobs can be succinctly captured as a 2D velocity corresponding to their difference in [width, height].

The self-supervised framework makes use of various loss terms that ensure the latent space is a well-formed / geometrically consistent metric space. The experimental evaluation shows that these methods successfully captures relationships across a variety of image and audio domains, better than standard dimensionality reduction baselines. When fed into a synthetic grid cell model, the learned upstream velocity signals result in much more grid-like firing patterns.

**Strengths:**

**Significance:** The work addresses an important problem in reconciling how grid cells could be involved in both spatial and abstract tasks. This is relevant for both the neuroscience community interested in grid cells / integrators, and potentially the machine learning community interested in transfer / generalization.

**Novelty:** The algorithm builds on ideas from SSL regarding prediction and promoting locally linear / geometrically consistent latent spaces. The perspective of abstract velocities is interesting and differently motivated than typical SSL work.

**Technical Quality:** The qualitative results are convincing, with the learned velocities accurately map their abstract domains. The model is also elegantly simple, yet can be applied to a variety of modalities.

**Presentation Clarity:** The writing is excellent, as is the figure design. The font for annotations in the figures (e.g. Figure 4 var/axes, Figure 3 isotropy equation) could be larger, but no major revisions suggested from me.

**Weaknesses:**

**Technical Quality**

1. **Dimensionality reduction:** Line 74. Is this a dimensionality reduction method? Can a single image be compressed into a low dimensional repr like the baselines that were compared to (e.g. autoencoder, PCA)? It seems to me that only a _pair_ of images could be compressed into a low-dimensional repr of the velocity. If trying to pass a single image twice into the encoder, a well-trained velocity would be 0.
2. **Preservation of cell-cell relationships:** Line 78, Section 2.6. I didn't follow the logic of this claim of what the model is predicting. Wouldn't showing that cell-cell relationships are preserved require showing that the velocity signals from multiple tasks can be fed to the same grid cell model simultaneously and use the same grid? Only one task is shown. Or is this claim presupposed by the grid cell model? In which case it wouldn't be a prediction.
3. **Orthogonality of velocities in triplet:** Line 163. The model computes velocity v12 from (i1, i2), then regresses g(i2, v12) to i3. But the actual velocity v23 between i2 and i3 could be quite different from v12, and in the worst case it is orthogonal. So would this introduce a systematic error? Or maybe its an assumption that velocity doesn't change too quickly along a path, in which case this assumption should probably be made explicit.
4. **Missing limitations:** I didn't find limitations in the Discussion section as indicated in the Checklist. Some limitations in particular that I'd like to see addressed are:
    (a) The biological plausibility of the proposed method, especially if this is meant to describe brain mechanisms for generating abstract velocities.
    (b) How naturalistic is the data collection process that requires closed-loops of random walks?

**Presentation Clarity**

5. **Missing grid cell model details:** Line 254, Is the grid cell model described in the paper/appendix? I couldn't find it or a reference to another work.
6. **Missing data generation details:** Line 153, It will be important to have these details on how the synthetic data was generated.
7. **Network vs circuit model:** Line 62 and elsewhere, I would suggest you rename this a "neural network" model, not a "neural circuit" model. The proposed SSL architecture is a deep learning style model. Typically, I think it's best practice to use "circuit" only if you are willing to provide specific commitments between parts of your model and the underlying neural circuit mechanisms in the nervous system. Meanwhile, "network" is less strict.
8. **Missing section number:** Line 207, Experimental Results should be differently numbered than Methods.

**Questions:**

See above.

**Limitations:**

I didn't find limitations in the Discussion section as indicated in the Checklist. See above for particular questions.

---

> ### Author Rebuttal · Authors · 2024-08-07
>
> We would like to thank the reviewer for their detailed feedback and for raising excellent questions. We will try to address all questions one-by-one.
>
> > **Figure 4 var/axes and Figure 3 isotropy equation**
>
> Thank you for the suggestions! We will make the necessary edits to the figure.
>
> > **Dimensionality reduction**
>
> Thank you for raising this point. Yes, our method is indeed a dimensionality reduction method. Our model estimates low-dimensional velocities between high-dimensional states, which can then be integrated to determine the low-dimensional representation of a given state. Dimensionality reduction methods typically apply over an ensemble of states, with the low-dimensional representation of a single state examined in relation to the low-dimensional representation of other states. Our method directly constructs these relative relationships by extracting low-dimensional velocities. As an example, we have generated a new figure (Fig. R3 in the attached PDF), showing states generated from a trajectory in the Moving Blobs environment. Through integration of our model-outputted velocities, the high-dimensional states within this trajectory fall on a plane (consistent with the data, since it can be minimally described through transitions in $\mathbb{R}^2$). PCA of the same set of states fails to capture this low-dimensional description, with ~24 principal components necessary to capture >95% of the variance.
>
> > **Preservation of cell-cell relationships**
>
> The key prediction of our model is that the cell-cell relationship between two cells is preserved across task modalities and across abstract cognitive domains. To simplify our explanation, we consider a grid cell model with a single grid module. Preserving cell-cell relationships does not require the model to _simultaneously_ receive inputs into the same grid module from multiple tasks. The only requirement is that the estimated velocities from each task are fed into the module, but not at the same time.
> The question of cell-cell relationships is whether, if two cells fire in an overlapping way in one mapped space/domain, they continue to do so in other domains; and if they fire in a non-overlapping way in one domain, whether they continue to be non-overlapping in all other domains (cf. Yoon et al. 2013 Nat Neuroscience). Our hypothesis is that the brain generates velocity signals from these different domains and feeds these signals into a fixed, prestructured grid cell circuit. Following the structure of the continuous attractor model assumed for the grid cells, we observe that if two grid cells are co-active for one task, they remain co-active for all other tasks. While this claim relies on the underlying grid cell model, it is not presupposed by it: preservation of cell-cell relationships follows from our hypothesis that low-dimensional velocities can be extracted from abstract domains, and thus the same continuous-attractor-based grid module can perform integration across domains.
>
>
> > **Orthogonality of velocities in triplet**
>
> As discussed briefly in L127-129, we assume that $v_{23} = v_{12}$ and ask the model to predict an $i_3$ solely to demonstrate that the model does not memorize the image features of $i_2$ through some encoding in the predicted velocity / latent space. We will clarify this in our paper. We demonstrate that this assumption is not necessary for model training in Fig. R2 in the attached PDF. Here, we retrain our network on the Moving Blobs task with trajectories that vary in velocity randomly at each time-step. Here we only predict an estimated $\hat{v}$ from $i_1$ and $i_2$, and then predict $i_2$ from $\hat{v}$ and $i_1$. We show that qualitatively and quantitatively, the obtained results remain very similar to the training paradigm we previously employed.
>
>
> > **Missing limitations: the biological plausibility of the proposed method**
>
> This is a great question. We believe our core loss framework is biologically plausible. The two critical losses we propose are the next-state prediction loss and the loop-closure loss: (1) The next-state prediction loss may occur through sensory systems computing an error that describes the difference between true and predicted sensory input; (2) The loop-closure loss can be computed through a neural integrator circuit such as the grid cell system. Experimental results show that grid cells process these abstract domains, so it is plausible that they provide some error / feedback towards the accurate mapping / organization of these spaces. We include further discussion related to this plausibility in text.
>
> In addition to the critical losses, we also used two auxiliary losses. The biological plausibility of these losses is not as clear as the critical loss terms; however, we note that they primarily aided in refinement of the obtained representations and were not crucial to our results (as demonstrated in our ablation results). We also fully agree that the optimization process we use (stochastic gradient descent and backpropagation) is not a process supported by biology. We will add these limitations to our discussion section.

---

> ### Author Response · Authors · 2024-08-07
> **Rebuttal, Part 2**
>
> > **How naturalistic is the data collection process that requires closed-loops of random walks?**
>
> Thank you for raising this question.
>
> While our models were trained with data consisting of loops, this training paradigm was for convenience (L179-181) and is not a necessity of the model. If arbitrary random walks were selected instead for training samples, self-intersections would automatically lead to loops within sub-sequences of the walks (as detected by $k<m$ with $i_k=i_m$) — these loops could then be used for the loop-closure loss, with all other losses applied on all loop and non-loop trajectories. As a simple proof of concept, we generated a dataset of arbitrary trajectories with 50% of the trajectories as loops and the remainder as non-loops. We applied loop closure loss on trajectories that formed loops and all other losses on the entire dataset of trajectories. Fig. R1 in the attached PDF demonstrates this paradigm for the 2D Stretchy Bird environment, where we continue to get consistent velocity representations with low transformation error to the true velocity distribution.
>
> In greater generality, we also note that the loop closure loss can also work on trajectories that are “almost loops” – i.e., a loss whose prefactor term scales with how close a trajectory is to forming a closed loop. However, running this analysis and tuning these parameters is beyond the scope of the current rebuttal period, and we will examine these results for the camera-ready version of our paper.
>
>
> > **Missing grid cell model details**
>
> Thank you for raising our attention to this. We provide here a brief description of the grid cell model used and will include a more detailed description in the appendix of the revised paper.
>
> We use an approximation of a continuous attractor model for a module of grid cells: we simulate patterned activity on a lattice of neurons as the sum of three plane waves, resulting in a hexagonal pattern of activity.  Input velocities are used to update the state of the activity on the lattice of neurons through updating the phases of the plane waves, leading to accurate integration of the input velocities.
>
> > **Missing data generation details**
>
> Thank you for raising our attention to this as well! We will provide additional details about the synthetic data generation in the appendix section. We briefly touched upon the training / testing set sizes and velocity distributions from which each synthetic environment was generated in the appendix, but we will add more details. We plan to include relevant details about the construction of the training / test set (e.g., how we generated random loop trajectories) and statistics about the environment states (e.g., size of image frames, maximum velocity allowed between consecutive states, etc.).
>
> > **Network vs. circuit model; Missing section number**
>
> Thank you for the excellent suggestions. We will change the terminology used to describe our model and fix the section numbering in our paper.

---

> ### Comment · Reviewer_sK54 · 2024-08-10
>
> Thank you for your rebuttal comments, additional results, and manuscript improvements!
>
> I urge you to clarify the "cell-cell relationships" discussion more in the paper, since the language can be a bit opaque, particularly for the non-grid-cell community. Maybe it would be good to discuss how this prediction could have been falsified (e.g. is it only if you weren't able to learn a good velocity signal to feed into the assumed grid cell model?). Also, in future work, it would be valuable to address the biological plausibility issues to make this a better model of the brain.
>
> Overall, your rebuttal has addressed my feedback, which did not raise major concerns. I am happy to maintain my original high score and support this paper's acceptance.

---

> > ### Author Response · Authors · 2024-08-11
> >
> > Thank you for the comments. We will certainly improve the discussion on grid cells and our predictions in the paper. We will ensure that the novelty of our work and predictions in the context of grid cells will be made more apparent to people outside the grid cell community.
> >
> > Discussion of how our predictions can be falsified is an excellent idea, and we will be sure to include this. To restate from earlier, we predict that if two grid cells are co-active for one task, they remain co-active for all other tasks. Further, if they fire in a non-overlapping way in one domain, they will continue to be non-overlapping in all other domains. This is critically dependent on the following being true: (1) that low-dimensional velocities can indeed be extracted from abstract domains, which we demonstrate through our learning framework, and (2) that the _same_ continuous-attractor-based grid module can perform integration across domains.
> >
> > This prediction may thus be falsified if the brain were to use _distinct_, independent grid cell modules to organize information from different modalities. It would also be falsified if the grid cells are not modeled by a pre-structured continuous attractor network which would function independently of the nature of the inputs. We will include and elaborate on the falsifiability of our predictions in the revised text.
> >
> > We are certainly interested in constructing more biologically plausible architectures and learning rules that can perform the computation described in our work, but as correctly noted, this would fall outside the scope of our current work.
> >
> > Thank you for your score and for supporting our paper's acceptance. We hope that you agree that our paper has improved, given your detailed feedback and suggestions. Please let us know if there are any additional questions that we may be able to address to consider a re-evaluated score.

---

### Author Rebuttal · Authors · 2024-08-07

We thank the reviewers for their careful and detailed review of our paper and their overwhelmingly positive feedback. In this general response, we address some common themes across the reviews. We will additionally provide detailed answers to each reviewer in subsequent comments. To go with this rebuttal, we provide a 1-page PDF with additional figures (labeled Fig. R1-R4).

All reviewers highlighted the importance and relevance of the problem, as well as the novelty of the proposed method. Reviewers also noted that our work was potentially impactful to both neuroscience and machine learning communities, found the model design to be simple yet elegant, the experiments and ablations to be thorough, and the writing and figure design to be excellent.

Two common critiques emerged across reviews: whether our training data consisting only of loops was justified, and whether our model was truly a/should be compared with dimensionality reduction method(s).

To address these issues and other questions raised by reviewers, we run new analyses in this rebuttal that we include in the attached PDF. Specifically:

1. To show that training purely on loops is not necessary, we find relatively unchanged results from our model when trained on data consisting of both random walks (non-loops) and random loops ($\textbf{Fig. R1}$). We provide more details in our response to $\textbf{sK54, SMXS, SEuj}$.
2. To further demonstrate that our model can truly be considered a dimensionality reduction method, we show the low dimensional representations of an ensemble of states from our 2D Moving Blobs environments. The low-dimensional representations from our model occupy a 2D subspace, reflecting the true data structure, unlike a PCA embedding that we show for comparison ($\textbf{Fig. R3}$). We provide more details in our response to $\textbf{sK54, SMXS, AXig}$.
3. To show how our model deals with boundary effects, we analyzed a trained model on a 2D Stretchy Bird task. Interestingly, the model’s encoder can continue to generate state-independent velocities, but the model’s decoder is state dependent and does not produce state predictions that lie beyond the implicit boundaries imposed by the training data ($\textbf{Fig. R4}$). We provide more details in our response to $\textbf{SEuj}$.

We sincerely thank the reviewers for their extensive feedback and believe that our manuscript has significantly improved as a result. We hope you agree that our proposed revisions and new analyses have enhanced our work and that our research can lead to a positive impact on both the neuroscience and machine learning communities.

---

### Decision · Program_Chairs · 2024-09-25

**Decision:**

Accept (poster)

**Comment:**

The present study considers a self-supervised learning paradigm to extend grid cells from the spatial domain to abstract domains. This is an important extension because we know grid cells need to represent abstract knowledge, while the spatial location is only a convenient metric that can be easily measured in lab experiments. Overall, the paper is conceptually novel and technically solid, and thus I recommend to be accepted. Please integrate reviewers' suggestions in the revised manuscript.